# R³DAO: Reactive Recovery and Reconstruction for Long-horizon Data Agent Orchestration

Quanxin Liu [1]   Rui Hao [1]   Ruida Xu [1]   Jianwei Zhong [1]   Changhu Chen [2]   Yijun Mo [1]

## Abstract

End-to-end data science agent workflows involve tightly coupled sub-processes with strong dynamic dependencies, posing a challenging long-horizon orchestration problem. Existing frameworks primarily rely on static, chain-like execution plans, which are prone to error propagation from early stages—often causing reasoning chain collapse and task failure, resulting in fragile inference and poor cost-effectiveness. To address these issues, we propose R³DAO, a reactive data agent orchestration framework based on feedback-driven topology evolution, aiming to build a dynamic evolutionary closed-loop of "hierarchical exploration, iterative recovery, and empirical convergence." First, we introduce a dynamic hierarchical task network that recursively decomposes global intent into macro-logical anchors and micro-operators, enabling low-cost exploration through dimensionality reduction in the logical space. Second, we establish a reactive topology reconfiguration mechanism that leverages semantic reflection to map execution anomalies into diagnostic signals, replacing costly global resets with localized topological optimization for resilient self-healing. Finally, semantic experience distillation implements a dual-loop accumulation that compresses long-horizon trajectories into structured prior, steering execution efficiency toward the optimal regime. Evaluations on the MLE-bench show that R³DAO achieves a 77.36% improvement in success rate over advanced R&D-Agent while maintaining competitive task scores. Notably, R³DAO compresses the average execution time by $36\times$ and limits token consumption to

just 104k per task, showcasing superior reliability, efficiency, and cost-effectiveness.

## 1. Introduction

End-to-end data science (DS) agents have demonstrated significant progress in automated orchestration (Feurer et al., 2022; Zoph & Le, 2016; Falkner et al., 2018), aiming to minimize human intervention in complex data analytical workflows (Sun et al., 2025a; Zheng et al., 2025; Liu et al., 2025; Gao et al., 2023). A complete DS task is not a collection of isolated operations but a sequence of highly coupled processes, ranging from data cleaning and feature extraction to joint model optimization (Zhang et al., 2023; 2024b). These stages are characterized by multi-level nested sub-processes, forming what is known as a long-horizon orchestration problem (Zheng et al., 2021; Tang et al., 2025). Consequently, the robustness of the global pipeline is predicated on the atomic precision of every underlying micro-operator.

However, the rigid dependencies inherent in long-horizon orchestration create a critical vulnerability (Fan et al., 2024). Minor errors, such as data type or dimension mismatches, can trigger cascading failures (Erdogan et al., 2025), rendering successful execution paths mathematically rare (Zhang et al., 2025). This places static orchestration in a dilemma: it is either too brittle to survive perturbations or prohibitively expensive to repair, given the high token cost of global resets.

To mitigate these issues, current strategies typically follow two conflicting paradigms. On one hand, some prioritize reasoning depth through exhaustive search or intensive multi-agent collaboration (Seo et al., 2025; Gandhi et al., 2025; Jiang et al., 2026; Giusti et al., 2025). While these methods improve the probability of identifying valid execution paths, they incur prohibitive computational overhead and latency, precluding agile deployment. Conversely, lightweight solutions prioritize efficiency through heuristic retries but often lack the semantic awareness to rectify deep logical contradictions (Sun et al., 2025b), leading to "silent failures" where erroneous states propagate unchecked across the orchestration horizon. This dichotomy reveals a critical gap: the need for a framework that reconciles the resilience

[1]School of Computer Science and Technology, Huazhong University of Science and Technology, Wuhan, China [2]Wuhan Research Institute of Posts and Telecommunications, Wuhan, China. Correspondence to: Rui Hao <d202581896@hust.edu.cn>, Yijun Mo <moyj@hust.edu.cn>.

*Proceedings of the 43rd International Conference on Machine Learning*, Seoul, South Korea. PMLR 306, 2026. Copyright 2026 by the author(s).

of search-based methods without sacrificing operational efficiency.

To bridge this gap, we propose R$^3$DAO, a reactive data agent orchestration framework driven by feedback-triggered topological evolution. R$^3$DAO is designed to construct a dynamic evolutionary closed-loop of "hierarchical exploration, iterative recovery, and empirical convergence." Our core contributions are summarized as follows:

- **Dynamic Hierarchical Task Network (D-HTN)**: We introduce D-HTN to recursively decouple global intentions into macro-logical anchors and micro-operators. By implementing logic-layer dimensionality reduction, the framework enables low-overhead exploration and fast trial-and-error.

- **Reactive Topology Reconfiguration (RTR)**: We establish the RTR mechanism to resolve orchestration failures. By utilizing semantic reflection to map physical errors into diagnostic signals, RTR facilitates resilient self-healing through local topological injections—such as *Refine*, *Insert*, or *Skip*—thereby enhancing the success rate without expensive global resets.

- **Semantic Experience Distillation**: To prevent reasoning overhead explosion, we implement a dual-loop accumulation mechanism that distills search trajectories into structured priors. This drives execution efficiency toward an optimal operational regime, significantly reducing token consumption.

We evaluate R$^3$DAO using a selection of 17 diverse competitions from the MLE-bench (Chan et al., 2025). These tasks are strategically chosen to span Low, Medium, and High complexity gradients, ensuring a comprehensive assessment of the framework's adaptability across heterogeneous data structures and long-range orchestration requirements. Experimental results reveal that R$^3$DAO significantly outperforms the advanced R&D-Agent, achieving a 77.36% improvement in task success rate. In terms of operational efficiency, R$^3$DAO compresses average execution time from 24 hours to under one hour. Furthermore, R$^3$DAO maintains a lean interaction footprint with an average token cost of only 104k per task, demonstrating that R$^3$DAO effectively reconciles reasoning resilience with extreme operational efficiency.

## Conflict of Interest Disclosure

The authors declare that they have no conflict of interest.

## 2. Related Work

Current autonomous DS orchestration generally fall into a dichotomy between resource-intensive exploration and frag-

ile heuristic execution. We categorize these developments into three primary streams:

**LLMs Agents for Data Science.** Large Language Models (LLMs) have evolved from standalone code generation to integrated task execution (Zhang et al., 2024a). General frameworks like AutoGen (Wu et al., 2024) and MetaGPT (Hong et al., 2024) leverage collaborative intelligence, while specialized DS agents such as DS-Agent (Guo et al., 2024) and Agent K (Bou-Ammar et al., 2025) focus on domain-specific analytical tasks. Despite their proficiency, these systems often struggle with multimodal integration and lack robust mechanisms to rectify structural errors (Jing et al., 2024). Crucially, without the ability to diagnose deep logical inconsistencies, they remain susceptible to error propagation in long-horizon workflows.

**AutoML and Specialized Search Frameworks.** Automated Machine Learning (AutoML) (Salehin et al., 2024; Trirat et al., 2024) has transitioned from static search methods Auto-WEKA (Thornton et al., 2013) and TPOT to meta-learning techniques such as Auto-Sklearn (Feurer et al., 2015). While effective, these systems are typically constrained by static search spaces. To bridge this, specialized agents such as AutoKaggle (Li et al., 2024) and R&D-Agent (Yang et al., 2025b) employ multi-agent collaboration, while search-driven frameworks like AIDE (Jiang et al., 2025) and SELA (Chi et al., 2024) utilize tree-search or evolutionary strategies. Although achieving high success rates, these methods incur prohibitive computational costs and latency—often requiring tens of hours per task —rendering them impractical for agile, resource-constrained deployment.

**Planning Structures and Context Efficiency.** Structured planning paradigms, evolving from CoT (Wei et al., 2022) to complex architectures like ToT (Yao et al., 2023) and GoT (Besta et al., 2024), aim to maintain reasoning coherence. However, linear context aggregation in these frameworks often leads to information saturation over long trajectories. Lightweight planners like Data Interpreter (Hong et al., 2025) prioritize speed but rely on superficial retry mechanisms, leaving them susceptible to silent failures where logical errors propagate undetected. While recent advances have begun exploring runtime structure adaptation—such as modularized workflow automation (Niu et al., 2025), dynamic multi-agent elimination (Wang et al., 2025b), and notebook-centric adaptive execution (You et al., 2025)—they primarily operate on flat graph routing or macro-level agent variations. This leaves a critical gap: the lack of a framework that can dynamically reconfigure intricate execution topologies based on fine-grained physical feedback while strictly preserving multi-level hierarchical goal integrity in long-horizon workflows.

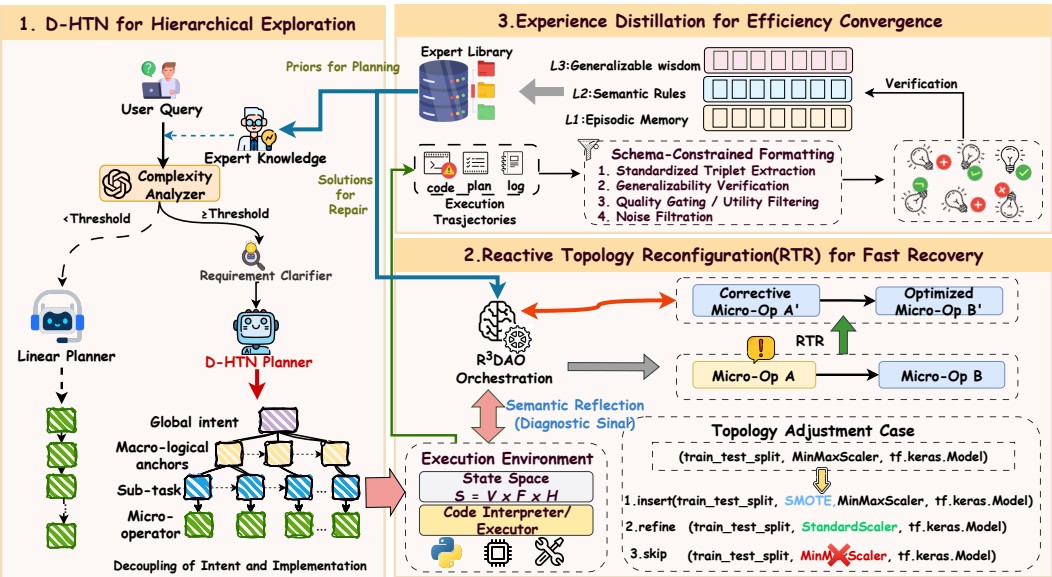

*Figure 1.* **The framework of R$^3$DAO.** The system constructs a dynamic closed-loop of "hierarchical exploration, iterative recovery, and empirical convergence". It leverages D-HTN to recursively decouple global intentions into macro-logical anchors and micro-operators. For execution anomalies, RTR utilizes semantic reflection to perform localized topological self-healing (Refine, Insert, or Skip). Finally, a dual-loop experience distillation mechanism compresses trajectories into structured priors to optimize future execution.

*Table 1.* Architectural comparison between R$^3$DAO and SOTA baselines.The comparison highlights differences in planning topology

| Method | Planning Topology | Reflective Mechanism | Experience Memory |
|---|---|---|---|
| OPENHANDS(Wang et al., 2025a) | Linear (ReAct) | ✗ | ✗ |
| MLAB(Huang et al., 2023) | Linear Chain | ✗ | ✗ |
| DATAINTERPRETER(Hong et al., 2025) | Dynamic Chain | Local Retry (Rule-based) | ✗ |
| AIDE(Jiang et al., 2025) | Tree Search | Best-of-N Selection | ✗ |
| R&D-Agent(Yang et al., 2025b) | Multi-chain Fusion | Uncontrollable | ✗ |
| *R$^3$DAO* **(Proposed)** | **D-HTN (Hierarchical)** | **RTR (Reactive)** | ✓**(Dual-loop)** |

## 3. Problem Formulation

Autonomous DS orchestration is formalized as a sequential decision-making process mapping a high-dimensional intention $I$ to a sequence of micro-operators $\sigma$ within a dynamic environment $\mathcal{S}$. Unlike general code generation, DS tasks (encompassing data cleaning, feature engineering, and model optimization etc.) involve stringent hard dependencies, where preceding outputs strictly constrain the subsequent feasible action space.

We define the execution environment as a dynamically evolving state space $\mathcal{S}$, structured as the Cartesian product of three critical dimensions:

$$\mathcal{S} = \mathcal{V} \times \mathcal{F} \times \mathcal{H} \qquad (1)$$

where $\mathcal{V}$ denotes the variable space ensuring metadata and memory-object continuity. $\mathcal{F}$ represents the file space managing persistent assets across the planning horizon. $\mathcal{H}$ indicates the environment space capturing library dependencies and system configurations. Within this space, an agent's behavior is modeled as a state transition process

$s_{t+1} = \Delta(s_t, a_t)$ driven by a deterministic code interpreter. Crucially, each transition must be verified through real-time physical feedback from the kernel rather than assumed priors.

To navigate the complexity of $\mathcal{S}$, we decouple operators into a hierarchical structure.

- **Macro-operators (Parent Tasks $pt_i$):** Logical milestones that define the workflow's topological skeleton (e.g., "Iterative Feature Selection"). When the Micro-Planner determines that a parent task requires multi-step expansion (e.g., split=true), it may first be decomposed into intermediate logical subtasks (the blue nodes in Figure 1).

- **Micro-operators (Atomic Nodes $ct_{i,j}$):** Executable leaf-level instantiations (e.g., specific Python code blocks) responsible for the underlying state transformations. Intermediate subtasks are eventually instantiated into these micro-operators.

Here, $i$ indexes the parent task $pt_i$, and $j$ indexes the $j$-th micro-operator $ct_{i,j}$ under that parent task. To distinguish hierarchical task indices from physical execution order, we use $t$ to denote the global execution step. The mapping between them is given by

$$t = \sum_{k<i} m_k + j \qquad (2)$$

where $m_k$ is the number of micro-operators instantiated under parent task $pt_k$. Thus, a single parent task may span multiple execution steps, and $t$ should not be conflated with the parent-task index $i$. Correspondingly, the execution-layer action $a_t$ denotes the physical state transition at step $t$, whereas the recovery action $a^{rtr} \in \{Refine, Insert, Skip\}$ operates at the planning layer to repair the task graph.

This hierarchy reflects DS engineering logic and provides the structural basis for the reactive evolution mechanism proposed in this work.

Task complexity is quantified by the number of micro-steps $n$ and dependency depth. In long-horizon orchestration ($n > 10$), the global success rate $P_{total}$ follows the multiplication law of conditional probabilities:

$$P_{total} = \prod_{i=1}^{n} p_i(s_t, a_t) \tag{3}$$

where $p_i$ is the probability of successful execution at step $i$ given the cumulative state $s_t$. Due to cascading dependencies, a localized deviation (e.g., a data drift causing $p_i \to 0$) inevitably triggers a chain failure, rendering the remaining inference path obsolete. This combinatorial sparsity explains the fragility of static planning: valid execution paths are exponentially sparse within the total search manifold. R³DAO leverages semantic reflection to detect these deviations and performs reactive topology reconfiguration to restore the $p_i$ distribution, ensuring task closure.

## 4. Methodology

R³DAO transforms fragile linear planning into a resilient, feedback-triggered topological evolution. By constructing a dynamic closed-loop of "hierarchical exploration, iterative recovery, and empirical convergence," the framework counteracts the combinatorial sparsity and cascading failures identified in Section 3. The comprehensive system architecture is illustrated in Figure 1, with key notations summarized in Appendix A.

### 4.1. D-HTN for Hierarchical Exploration

DS orchestration often faces a search space explosion, denoted as $O(b^n)$, where $b$ is the branching factor of micro-operators and $n$ is the sequence length. D-HTN mitigates this by implementing a logic-layer dimensionality reduction through a recursive decoupling protocol.

The decomposition operator $\Phi$ projects the global intention $I$ onto a macro-logical skeleton $\mathcal{G}_{macro}$ using domain expert library $\mathbb{G}$:

$$\Phi : (I, \mathbb{G}) \to \mathcal{G}_{macro} = pt_1, \dots, pt_k \tag{4}$$

By anchoring semantic milestones at the macro-level, the framework reduces the effective search depth from $n$ to $k$

---

**Algorithm 1** Reactive Topology Reconfiguration (RTR)

**Require:** Task graph $G_t$, anomaly $e$, state $s_t$, retry count $k$, max retries $N_{max}$
**Ensure:** Updated graph $G_{t+1}$
1: $\delta_{diag} \leftarrow \mathcal{R}(e, s_t, ct_{i,j})$
2: $a^{rtr}, r, \iota \leftarrow \delta_{diag}$
3: **if** $a^{rtr} == Refine$ **and** $k < N_{max}$ **then**
4:      $ct_{i,j}.code \leftarrow Refine(\iota)$
5:      **return** $G_t$
6: **else if** $a^{rtr} == Insert$ **then**
7:      $ct_{new} \leftarrow \text{GENNODE}(\iota)$
8:      $G_{t+1} \leftarrow Insert(G_t, ct_{new}, ct_{i,j})$
9:      **return** $G_{t+1}$
10: **else**
11:      $G_{t+1} \leftarrow \text{RECONSTRUCT}(G_t, \tau_{<t}, \mathcal{K}_t)$
12:      **return** $G_{t+1}$
13: **end if**

---

($k \ll n$), ensuring trajectories remain within a convergent expert-defined manifold and validating global logic with minimal overhead.

The instantiation operator $\Psi(pt_i, s_t)$ adaptively expands macro-logical into micro-operators $\{ct_{i,j}\}$ based on real-time observations of the physical state $s_t$, particularly the metadata distribution in the Variable Space $\mathcal{V}$.

$$\Psi : (pt_i, s_t) \to ct_{i,1}, \dots, ct_{i,m} \tag{5}$$

This collapse-and-expansion mechanism dynamically modulates observation density. When environmental uncertainty increases, $\Psi$ triggers fine-grained expansion to decompose complex logic into monitorable units, pre-empting risks via micro-deviation detection.

### 4.2. RTR for Fast Recovery

Execution exceptions $e$ in the physical environment represent a transition into a state of increased uncertainty within $\mathcal{S}$. RTR functions as a feedback-driven state corrector, utilizing semantic reflection to map physical feedback into structured topological adjustments to restore the execution probability distribution $p_i$ (Eq. 3). Upon micro-operator $ct_{i,j}$ failure, the reflection operator $\mathcal{R}$ analyzes the runtime anomaly together with the residual environment state to generate a diagnostic signal:

$$\mathcal{R} : (e, s_t, ct_{i,j}) \to \delta_{diag}, \quad \delta_{diag} = \langle a^{rtr}, r, \iota \rangle \tag{6}$$

where $a^{rtr} \in \{Refine, Insert, Skip\}$ denotes the recovery action, $r$ is the diagnosed root cause, and $\iota$ is the corresponding corrective instruction. In this way, raw execution anomalies are converted into topology-aware recovery signals rather than treated as generic retry events.

**Algorithm 2** Semantic Reflection Operator $\mathcal{R}$

---

**Require:** Anomaly $e_t$, state $s_t$, failed node $ct_{i,j}$, retry count $k$, max retries $N_{max}$
**Ensure:** Diagnostic signal $\delta_{diag} = \langle a^{rtr}, r, \iota \rangle$
1: $error\_type \leftarrow \text{CLASSIFYERROR}(e_t)$
2: **if** $error\_type \in \{SyntaxError, LogicError\}$ **and** $k < N_{max}$ **then**
3:    $\iota \leftarrow \text{REFINEINST}(ct_{i,j}, e_t, s_t)$
4:    **return** $\langle Refine, r, \iota \rangle$
5: **else if** $error\_type \in \{DimensionMismatch, MissingDependency\}$ **then**
6:    $\iota \leftarrow \text{INSERTNODE}(e_t, s_t, ct_{i,j})$
7:    **return** $\langle Insert, r, \iota \rangle$
8: **else**
9:    $\iota \leftarrow \text{RECONSTRUCT}(\tau_{<t}, \mathcal{K}_t)$
10:    **return** $\langle Skip, r, \iota \rangle$
11: **end if**

---

To improve reproducibility, we further make the routing logic of $\mathcal{R}$ explicit:

As detailed in Algorithm 1 and Algorithm 2, RTR restores logical consistency through three localized strategies:

- **Refine**: rectifies recoverable node-local faults in place while preserving the current topology.
- **Insert**: injects a compensatory node when failure is caused by a missing prerequisite or structural incompatibility.
- **Skip**: triggers bounded local reconstruction when node-level recovery is exhausted.

For example, in the TGS Salt segmentation task, a target-size mismatch ($101 \times 101$ vs. $96 \times 96$) is classified as a structural incompatibility rather than a node-local error, and therefore triggers *Insert* instead of repeated refinement. This formulation also provides a scoped analytical view of RTR. While static linear planning suffers multiplicative success decay with horizon ($P_{\text{total}} = \prod_{i=1}^{n} p_i$), RTR separates node-local refinement from structural recovery, explaining why *Insert* is necessary when failure is caused by a missing prerequisite rather than a recoverable local fault.

### 4.3. Experience Distillation for Efficiency Convergence

Frequent topological evolution ensures task completion but risks a reasoning overhead explosion in the absence of long-term memory. Semantic experience distillation compresses extensive trajectories into complexity-bounded priors. The distiller $\mathcal{E}$ projects high-dimensional historical feedback $\tau_{<t}$ onto structured knowledge $\mathcal{K}_t$:

$$\mathcal{K}t = \mathcal{E}(\tau_{<t}, s_t, e) \tag{7}$$

$\mathcal{E}$ extracts metadata invariants $\mathcal{V}_{meta}$ and file path constraints $\mathcal{F}_{path}$, compressing the context burden from $O(N)$ to a constant-scale $O(1)$. This prevents token overflow and ensures the planner operates in the most information-dense context, effectively minimizing redundant search steps while maximizing the probability of task completion.

To bridge the logic-physical gap during reconfiguration, the protocol $\Pi$ enforces state constraints from $\mathcal{K}_t$ onto new nodes.

$$s_{init}(ct_{new}) \leftarrow \Pi(s_{prev}, \mathcal{K}_t) \tag{8}$$

In the inner loop, $\mathcal{K}_t$ ensures immediate consistency for dynamic topology. In the outer loop, validated trajectories are persisted into the expert library $\mathbb{G}$.

## 5. Experiments

### 5.1. Experimental Setup

#### 5.1.1. BENCHMARK AND TASK TAXONOMY.

To evaluate the efficacy of R³DAO in data science workflows, we employ MLE-Bench (Chan et al., 2025), a high-fidelity benchmark for end-to-end machine learning engineering. We curate a representative subset of 17 tasks spanning modalities such as Tabular, NLP, Vision, and Signal Processing, with data scales ranging from $< 10$MB to $> 30$GB. Based on structural complexity and execution depth, tasks are categorized into Low ($< 5$ steps), Medium (6–9 steps), and High ($> 10$ steps) complexity gradients. This taxonomy allows us to isolate the marginal contributions of D-HTN and RTR under varying degrees of uncertainty. The subset was selected according to these coverage criteria rather than preliminary outcomes; detailed task-level statistics are provided in Appendix B, while full-benchmark evaluation remains future work.

#### 5.1.2. BASELINES, METRICS, AND IMPLEMENTATION SETUP

We benchmark R³DAO against a hierarchy of agents, including specialized SOTA frameworks (*e.g.*, MLAB(Huang et al., 2023), R&D-Agent(Yang et al., 2025b)) and reproduced baselines (*e.g.*, AIDE(Jiang et al., 2025) with a 16-trajectory tree search (tree search with depth 8 and branching factor 2) and DATAINTERPRETER(Hong et al., 2025) (10-round horizon with 3 local retries per step). To provide a holistic assessment of prediction accuracy and execution robustness, we introduce the Weighted Comprehensive Score:

$$S_{total} = 0.6 \times S_{norm} + 0.4 \times S_{comp} \tag{9}$$

Here, $S_{norm}$ normalizes raw scores (*e.g.*, RMSE, Accuracy) to a $[0, 1]$ scale relative to the leaderboard's *Gold* and *Median* benchmarks, while $S_{comp}$ quantifies pipeline completion from 1.0 (Success) to 0.0 (Failure). Detailed formulas

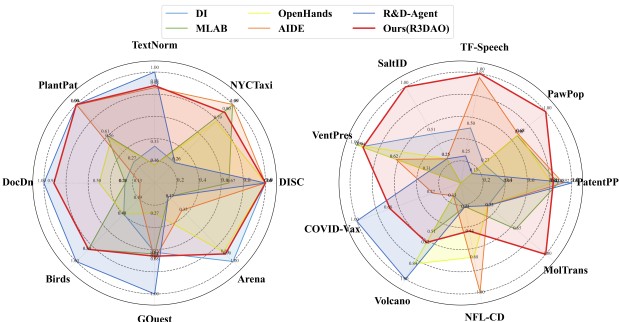

*Figure 2.* Comprehensive score distribution across 17 MLE-bench tasks. R³DAO (Red) exhibits significantly broader coverage than baselines. In high-complexity tasks such as *SaltID* and *MolTrans*, R³DAO maintains robust performance, whereas other frameworks (e.g., MLAB and OpenHands) frequently collapse due to execution failures.

and state definitions are provided in Appendix C. Each task is executed in 3 independent runs, with metrics averaged across runs; the inner-loop memory $K_t$ is reset per task, and task order is varied across runs to reduce ordering effects.

A critical distinction of our evaluation is the resource-constrained "Agile" setting. Unlike standard MLE-Bench protocols that utilize proprietary models (*e.g.*, GPT-4o (Hurst et al., 2024), GPT-5 (Georgiou, 2025)) under 24-hour limits and massive compute (36 vCPUs, 440GB RAM), we deploy R³DAO using the Qwen3-Max(Yang et al., 2025a) model on a single consumer-grade NVIDIA RTX 3090 GPU (24GB). Furthermore, we restrict the average execution time to < 1 hour per task. This setup is intentionally designed to demonstrate that R³DAO's gains stem from architectural superiority, such as efficient D-HTN planning and RTR-based self-healing, rather than brute-force computational scaling.

## 5.2. Main Results

Table 2 reports results across 17 tasks. We use concise task abbreviations (Appendix 7). The left panel provides a controlled intra-regime comparison among agile frameworks (Ours, AIDE, and DI) under the same model and < 1h budget, while the right panel reports high-resource references (MLAB, OpenHands, and R&D-Agent; GPT-4o/5, 12–24h) used only for cost-efficiency context rather than controlled architectural comparison.

### 5.2.1. COMPETITIVE PERFORMANCE DESPITE STRICTER EVALUATION

Remarkably, R³DAO-evaluated under far more constrained conditions-matches or exceeds the performance of systems reported on the official leaderboard, including the high-resource MLAB framework (GPT-4o, 24h). As illustrated in Figure 2, this advantage is most evident in complex, failure-prone workflows.

**Recovery from Execution Failures**: On *TF-Speech* and *COVID-Vax*, MLAB's leaderboard entry scores 0.0, indicating a complete inability to produce a valid submission within its allocated 24 hours. In contrast, R³DAO successfully navigates these tasks using its Reactive Topology Reconfiguration (RTR) mechanism, achieving scores of 0.532 and 0.679, respectively. This highlights the limitations of non-reactive planning in volatile environments.

**Efficient Multimodal Orchestration**: For the image denoising task *DocDn*, R³DAO attains a score of 0.910, vastly outperforming MLAB's reported 0.264 and approaching the performance of R&D-Agent (1.000)-despite using only Qwen3-Max and a one-hour runtime limit.

### 5.2.2. CONSISTENT GAINS IN RELIABILITY AND EFFICIENCY

Figure 3 illustrates the decoupling of performance intensity (Score) and execution robustness (Success Rate). R³DAO's architectural advantages are particularly evident in its ability to maintain high success rates where other agents fail.

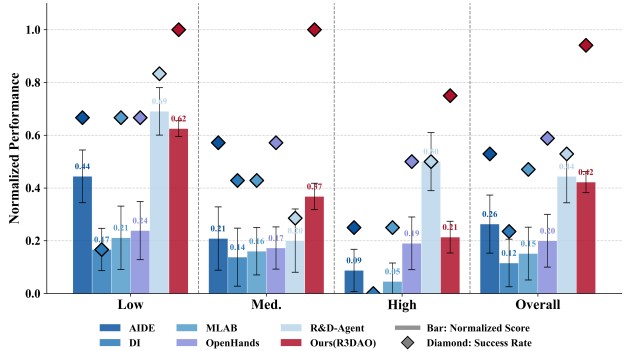

*Figure 3.* Performance & Success Rate Comparison across Complexity Levels. Bars denote Normalized Performance Scores, while ◇ indicate Task Success Rates (submission validity). R³DAO achieves a 100% Success Rate in Low and Medium complexity tasks. In High Complexity tasks, R³DAO maintains competitive score (0.21) while significantly outperforming open-source baselines (AIDE, OpenHands) in submission validity, validating the resilience of the RTR mechanism.

**SOTA Performance in Medium Complexity**: In the Medium Complexity bracket, R³DAO achieves the highest normalized score of 0.37, surpassing both the high-resource R&D-Agent and other agile baselines. Crucially, it maintains a perfect 1.0 Success Rate, indicating that the D-HTN planner effectively navigates coupled feature engineering tasks without logical collapse.

**Resilience in High-Horizon Tasks**: In the High Complexity regime (e.g., *Volcano*, *MolTrans*), extended dependency chains typically cause "silent failures" in baselines. While the GPT-5 based R&D-Agent leads in raw scoring capacity due to its stronger base model, R³DAO achieves a remark-

*Table 2.* Main Results on MLE-Bench subset. We compare R³DAO against baselines in two settings. **Left (Efficient Regime)**: Baselines reproduced in our constrained environment (Qwen3-Max, < 1h), including AIDE (Jiang et al., 2025), DataInterpreter (Hong et al., 2025), and R³DAO (Ours). **Right (High-Resource Regime)**: SOTA metrics cited from the official leaderboard (GPT-4o/5, 12–24h), including MLAB (Huang et al., 2023), OpenHands (Wang et al., 2025a), and R&D-Agent (Yang et al., 2025b). **Bold** and underline denote the best score in each group and the global best, respectively.

| Comp. | Task | Efficient Regime (Model: Qwen3-Max, Time: < 1h) | | | High-Resource Regime (Model: GPT-4o/5, Time: 12–24h) | | | Ours Impv. |
|---|---|---|---|---|---|---|---|---|
| | | AIDE | DI | R³DAO (Ours) | MLAB | OpenHands | R&D-Agent | vs. AIDE |
| **Low** | DISC | **1.000** | **1.000** | **1.000** | 0.670 | **1.000** | **1.000** | 0.0% |
| | NYCTaxi | **0.510** | 0.132 | 0.458 | 0.508 | 0.401 | 0.132 | -10.2% |
| | TextNorm | 0.694 | 0.268 | **0.709** | 0.132 | 0.132 | **0.807** | +2.1% |
| | PlantPat | 0.994 | 0.268 | **1.000** | 0.609 | 0.556 | **1.000** | +0.6% |
| | DocDn | 0.132 | 0.268 | **0.910** | 0.264 | 0.501 | **1.000** | **+589%** |
| | Birds | 0.132 | 0.268 | **0.573** | 0.573 | 0.268 | **0.678** | **+334%** |
| | *Average* | *0.577* | *0.367* | ***0.775*** | *0.459* | *0.476* | *0.770* | *+34.3%* |
| **Med** | GQuest | **0.668** | 0.609 | 0.643 | 0.624 | 0.268 | **0.975** | -3.7% |
| | Arena | 0.264 | **0.758** | 0.685 | 0.132 | 0.666 | 0.132 | -9.6% |
| | PatentPP | **0.614** | 0.268 | 0.555 | 0.585 | 0.544 | **0.665** | -9.6% |
| | PawPop | 0.680 | 0.268 | **1.000** | 0.664 | 0.683 | 0.132 | **+47.1%** |
| | TF-Speech | 0.514 | 0.268 | **0.532** | 0.000 | 0.000 | 0.132 | +3.5% |
| | SaltID | 0.132 | 0.268 | **0.524** | 0.132 | 0.132 | 0.132 | **+296%** |
| | VentPres | 0.268 | **0.412** | 0.405 | 0.268 | **0.431** | 0.132 | -1.6% |
| | *Average* | *0.449* | *0.407* | ***0.621*** | *0.344* | *0.389* | *0.329* | *+38.3%* |
| **High** | COVID-Vax | 0.268 | 0.000 | **0.679** | 0.000 | 0.000 | **1.000** | **+153%** |
| | Volcano | 0.132 | 0.132 | **0.618** | 0.509 | 0.843 | **1.000** | **+368%** |
| | NFL-CD | **0.610** | 0.132 | 0.268 | 0.132 | **0.414** | 0.132 | -56.0% |
| | MolTrans | 0.132 | 0.132 | **0.415** | 0.268 | 0.132 | 0.132 | **+214%** |
| | *Average* | *0.286* | *0.099* | ***0.495*** | *0.227* | *0.347* | *0.566* | *+73.1%* |
| *Grand Average* | | *0.456* | *0.321* | ***0.646*** | *0.357* | *0.410* | *0.540* | *+41.7%* |

able **Success Rate of 75%** (Diamond marker), significantly outperforming AIDE and OpenHands, which frequently fail to produce valid submissions. This confirms that the RTR mechanism effectively prevents error propagation, ensuring task closure even when model capacity is constrained.

**Overall Efficiency-Robustness Balance**: On the aggregated Overall metric, R³DAO secures a score of **0.42** with a near-perfect success rate. This positions it as the most reliable agile framework, offering a viable alternative to proprietary models by trading marginal raw performance for exceptional execution stability.

In total, R³DAO achieves the highest score among all *reproduced* agents in 10 out of 17 tasks and is the only system-among both reproduced and leaderboard entries-that successfully completes several high-stakes challenges where even GPT-4o-based agents fail entirely. This confirms R³DAO's capability as a reliable and scalable solution for real-world automated machine learning engineering. For transparency, we report sensitivity analyses and raw-score / medal-tier summaries in the appendix E, with error bars provided in Figure 3.

### 5.3. Efficiency and Cost-Effectiveness Analysis

The practical utility of autonomous agents depends on both performance and resource efficiency. We present a multi-

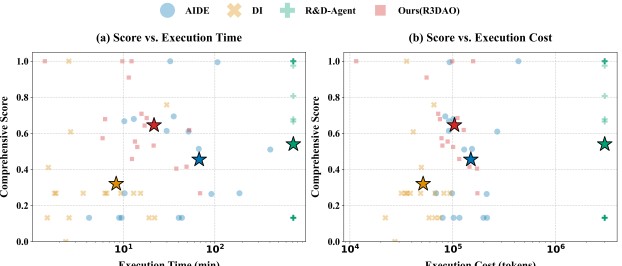

*Figure 4.* Performance-Cost Trade-off Scatter. (a) Comprehensive Score vs. Execution Time (log scale). (b) Comprehensive Score vs. Token Consumption (log scale). Large stars denote framework average performance, while semi-transparent markers indicate individual task results. R³DAO (red star) occupies the optimal top-left region in both plots, achieving SOTA-level performance with computational costs significantly lower than R&D-Agent (green star) and AIDE (blue star).

dimensional analysis of these factors in Table 3 and visualize the trade-offs in Figure 4.

**Efficacy Frontier Dominance.** As shown in Fig. 4a for time efficiency, R³DAO (Red Star) is positioned distinctly to the left of the high-performance cluster. While R&D-Agent (Green Star) achieves a marginal score advantage (+0.02), it resides in the $10^2 \sim 10^3$ minute range. In contrast, R³DAO clusters in the $10^1$ minute range, delivering comparable in-

*Table 3.* Efficiency and reliability metrics. **Time**: Average execution time per task (min). **Tokens**: Average token consumption. **Success Rate**: Percentage of valid submissions. R$^3$DAO matches the performance of the heavy R&D-Agent (0.42 vs. 0.44) while reducing costs by over **30×**.

| Model | Time (min) ↓ | Tokens ↓ | Score ↑ | Success Rate ↑ |
|---|---|---|---|---|
| DataInterpreter | **4.0** | **49k** | 0.12 | 0.24 |
| AIDE | 67.4 | 150k | 0.26 | 0.53 |
| MLAB | 1440 (24h) | – | 0.15 | 0.47 |
| OpenHands | 1440 (24h) | – | 0.20 | 0.59 |
| R&D-Agent | 720 (12h) | ∼3000k | **0.44** | 0.53 |
| **R$^3$DAO (Ours)** | 21.7 | 103k | 0.42 | **0.94** |

telligence in approximately 3% of the time. Token Economy in Fig. 4b shows that R$^3$DAO consumes only 103k tokens on average-roughly $\frac{1}{30}$th of the consumption reported by R&D-Agent (∼3000k). This positions R$^3$DAO as the only framework that breaks the "intelligence-cost" correlation, offering high-fidelity reasoning on consumer-grade budgets.

**Execution Reliability.** R$^3$DAO achieves a standout Success Rate of 0.94 in Table 3, significantly surpassing all baselines (0.24–0.59). While baselines like AIDE (0.53) frequently fail due to error propagation, RTR mechanism ensures resilient execution. This is visually corroborated in Figure 4, where R$^3$DAO's task markers (red squares) cluster tightly in the upper region, whereas baselines show high variance with many points collapsing to the bottom (low score).

**Conclusion on Efficiency.** Comparing to agile baselines, R$^3$DAO strikes an optimal balance. While DataInterpreter is faster (4.0 min), its low performance score (0.12) limits its use in complex reasoning. R$^3$DAO democratizes robust data science agents by enabling SOTA performance on standard hardware.

### 5.4. Ablation Study

*Table 4.* Overall Ablation Analysis (Average Across All Tasks). D-HTN is fundamental for reasoning depth (Score), RTR is critical for execution resilience (Success Rate), and Memory ensures efficiency (Tokens).

| Variant | Norm. Score | Success Rate | Avg. Tokens |
|---|---|---|---|
| **R$^3$DAO (Full Model)** | **0.422** | **94.1%** | **104 k** |
| w/o RTR (Static Retry) | 0.268 | 52.9% | 132 k |
| w/o Exp. Memory | 0.385 | 82.4% | 215 k |
| w/o D-HTN (Linear Plan) | 0.195 | 41.2% | 189 k |

To quantify the contribution of each component, we evaluate four variants across the entire 17-task benchmark (Table 4): (1) Full Model (R$^3$DAO); (2) w/o RTR, replacing topological self-healing with static retries; (3) w/o Exp. Memory, removing semantic distillation ; and (4) w/o D-HTN, replacing hierarchical planning with a linear Chain-of-Thought (CoT) planner.

**Impact of Reactive RTR.** Removing RTR causes a catastrophic drop in Success Rate from 94.1% to 52.9%. Without the ability to dynamically inject diagnostic nodes (e.g., *Insert/Refine*), the agent fails to recover from "chain failures" in long-horizon tasks. For instance, in *Volcano*, static retries succumb to repetitive memory overflows, whereas RTR pivots to a chunk-processing strategy to resolve the impasse. Table 5 further breaks RTR into its internal recovery modes, showing that *Refine* is more common in low-complexity tasks, whereas *Insert* becomes dominant as orchestration complexity increases.

*Table 5.* Distribution of RTR action types across task complexity levels (17 tasks, 3 runs each, 135 total interventions). *Insert* dominates high-complexity tasks (60.8%), *Refine* dominates low-complexity tasks (80.0%), and *Skip* remains rare overall (8.1%), indicating that RTR adapts its recovery mode to orchestration complexity.

| Complexity | RTR Calls | Refine | Insert | Skip |
|---|---|---|---|---|
| Low (6 tasks) | 20 | 80.0% (16) | 15.0% (3) | 5.0% (1) |
| Medium (7 tasks) | 64 | 42.2% (27) | 48.4% (31) | 9.4% (6) |
| High (4 tasks) | 51 | 27.5% (14) | 60.8% (31) | 11.8% (6) |
| Overall | 135 | 44.4% | 47.4% | 8.1% |

**Impact of Experience Distillation.** Ablating experience doubles the computational cost, with token consumption spiking by 106% (104k to 215k). This stems from the high overhead of tabula rasa exploration. Distilled experience serves as a critical shortcut, allowing the agent to prune the search space and bypass redundant experiments to converge directly on valid pipelines.

**Impact of D-HTN Planning.** Replacing D-HTN with linear planning yields the lowest Normalized Score (0.195). Linear chains suffer from severe context drift in long horizons (> 10 steps), losing track of global optimization goals. This highlights that hierarchical decomposition is a structural necessity for complex reasoning depth.

### 5.5. Case Study: Self-Healing in Action

Figure 5 depicts a recovery trajectory in the *TGS Salt Identification* task. Standard U-Net architectures often fail due to $101 \times 101$ spatial dimension mismatches. Unlike "blind retries" in other agents, R$^3$DAO retrieves spatial shrinkage priors and triggers RTR to reconfigure the network topology. By injecting corrective padding and upsampling layers rather than merely tuning parameters, R$^3$DAO achieves a structural evolution that addresses the root architectural flaw.

## 6. Conclusion and Future Work

This paper presented **R$^3$DAO**, a reactive orchestration framework addressing the fragility of autonomous DS agents in long-horizon orchestration. By integrating D-HTN for structured exploration, RTR mechanism for self-healing,

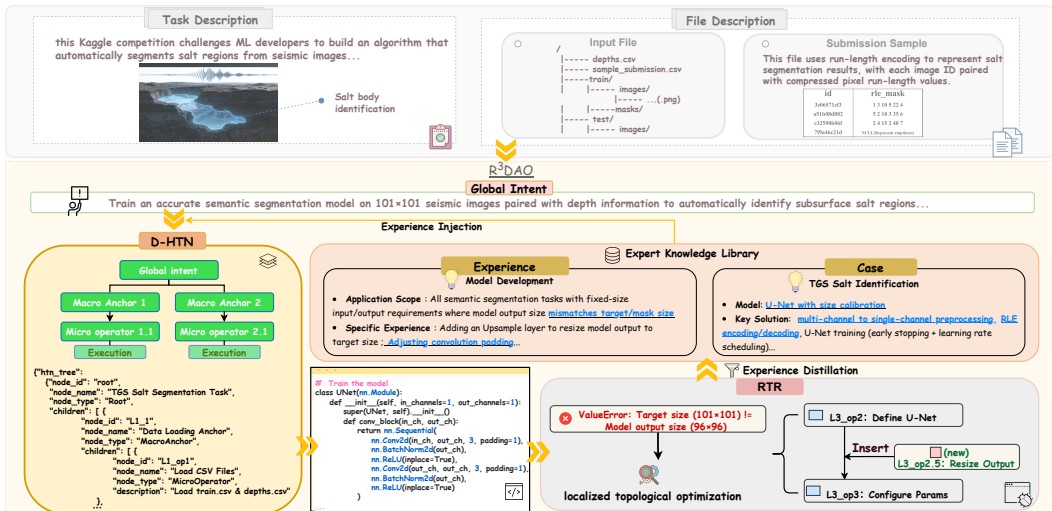

Figure 5. **Case Study on TGS Salt Identification.** The trajectory illustrates the self-healing process of $R^3$DAO. Upon detecting an execution anomaly, the RTR mechanism performs localized topological optimization to automatically resolve architectural flaws and ensure pipeline closure.

and Dual-loop Experience Distillation for knowledge reuse, R³DAO successfully bridges the gap between static planning and dynamic execution environments.

Extensive evaluation on MLE-Bench demonstrated that $R^3$DAO achieves state-of-the-art performance under strict efficiency constraints-running on a single consumer GPU with an average task runtime of less than 1h. Notably, it matches or exceeds the performance of top leaderboard entries such as R&D-Agent, which rely on significantly more powerful models and 12h execution budgets. This demonstrates that architectural resilience can effectively compensate for limited model capacity, providing a scalable and robust foundation for autonomous data science in complex, real-world analytical workflows.

**Limitations.** Despite its strengths, $R^3$ DAO faces several limitations. First, the RTR mechanism assumes that execution failures can be diagnosed and repaired using contextual cues from the environment and the LLM's internal knowledge. In scenarios where critical procedural knowledge is diluted by long context windows or polluted by outdated information, the agent may fail to generate precise repair instructions, potentially leading to redundant retry loops. Second, while R³DAO supports multimodal tasks via external tools, it lacks intrinsic visual reasoning capabilities. Complex data interpretation—such as inferring trends from multi-axis plots—remains a challenge compared to frameworks with native multimodal integration. Third, our analysis is intentionally scoped: it explains the multiplicative collapse of static long-horizon planning and the distinction between node-level refinement and topology-level insertion, but does not claim convergence to a globally correct solution or a full formalization of LLM decision quality in realistic environments.

**Future Work.** Future work will prioritize two directions. First, we will integrate Reinforcement Learning from Execution Feedback (RLEF) to transition topological evolution from heuristic reflection to optimized, data-driven policies. Second, we plan to extend $R^3$DAO to multi-agent settings for socialized topology evolution. In this paradigm, specialized agents can perform architectural injections—such as dynamically deploying monitoring agents—to stabilize long-horizon orchestration under extreme environmental volatility.

## Acknowledgements

This work was supported by the National Science and Technology Major Project under Grant No. 2026ZD1307000.

## Impact Statement

This paper presents an autonomous agent framework capable of executing complex data science workflows. The primary goal of this research is to assist data scientists by automating routine and error-prone tasks (e.g., data cleaning, baseline modeling), thereby allowing human experts to focus on high-level hypothesis formulation and decision-making.

However, we acknowledge potential societal risks. As autonomous agents become more proficient, there is a concern regarding the displacement of entry-level data analysis roles. Furthermore, automated code generation carries inherent

security risks; if deployed without sandboxing, agents could inadvertently execute malicious code or leak sensitive data. We strongly advocate for the deployment of such agents within strictly sandboxed environments (e.g., Docker containers, as used in our experiments) and emphasize that human oversight remains essential for critical decision-making processes.

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

## A. Summary of Key Notations

*Table 6.* Summary of Key Notations used in the R$^3$DAO formalization.

| Symbol | Definition | Description |
|---|---|---|
| $\mathcal{S}$ | Global State Space | Cartesian product of subspaces $\mathcal{V} \times \mathcal{F} \times \mathcal{H}$ representing the full environment. |
| $\mathcal{V}, \mathcal{F}, \mathcal{H}$ | Subspaces | Variable Space, File Space, and Environment configuration Space. |
| $s_t$ | State at step $t$ | A discrete snapshot of the environment at logical time step $t$. |
| $\mathcal{A}$ | Action Space | The set of all valid micro-operators executable in the domain. |
| $\Delta$ | Transition Function | Deterministic mapping $\mathcal{S} \times \mathcal{A} \to \mathcal{S}$ via code execution. |
| $I$ | Global Intention | High-dimensional natural language requirement from the user. |
| $G$ | Task Graph | Directed Acyclic Graph representing the workflow topology. |
| $\Phi$ | Decomposition Op. | Macro-level mapping: $I \to \{pt_1, \ldots, pt_k\}$. |
| $pt_i$ | Parent Task | Macro-logical anchor serving as a task milestone. |
| $\Psi$ | Instantiation Op. | State-aware expansion: $pt_i \to \{ct_{i,j}\}$. |
| $ct_{i,j}$ | Micro-operator | Atomic execution node (indivisible code block). |
| $\mathcal{R}$ | Semantic Reflection | Mapping $(e, s_t)$ to diagnostic signal $\delta_{diag}$. |
| $\delta_{diag}$ | Diagnostic Signal | Structured triple $\langle a, r, \iota \rangle$ for recovery guidance. |
| $\mathcal{K}_t$ | Knowledge Prior | Distilled experience used for pruning and consistency. |
| $\mathcal{E}$ | Experience Distiller | Trajectory compression function: $\tau_{<t} \to \mathcal{K}_t$. |
| $\Pi$ | Alignment Protocol | State projection ensuring logic-physical consistency. |

## B. Benchmark Details

Table 7 presents the comprehensive specifications for the 17 curated MLE-Bench tasks used in our experiments. To mitigate potential selection bias and ensure a representative evaluation independent of preliminary outcomes, these tasks are strategically selected based on three explicit criteria: (1) *Modality coverage*, spanning Tabular, NLP, Vision, Audio, and Signal Processing to prevent data-type dominance; (2) *Complexity gradient*, covering Low ($< 5$ steps), Medium (6–9 steps), and High ($> 10$ steps) to span the full orchestration-depth spectrum; and (3) *Data scale*, varying from $< 10$MB to $> 30$GB to balance lightweight and heavy-compute workflows. This table serves as both a mapping reference for the abbreviations used in the main text and a detailed statistical summary of the dataset characteristics.

## C. Evaluation Metrics Details

**Normalized Performance Score ($S_{norm}$)** We define this metric to handle the diverse metrics (e.g., Accuracy, RMSE, F1) used across MLE-Bench tasks. Since these metrics have varying scales and optimization directions, we normalize the raw score $x$ based on the task-specific Leaderboard *Gold* and *Median* thresholds. We map all metrics to a unified $[0, 1]$ scale where higher is better:

**Case 1: Metric Type = Max (Higher is Better).**

$$S_{norm} = \begin{cases} 1.0 & \text{if } x \geq \text{Gold} \\ 0.5 + 0.5 \cdot \frac{x - \text{Median}}{\text{Gold} - \text{Median}} & \text{if Median} \leq x < \text{Gold} \\ 0.5 \cdot \frac{x}{\text{Median}} & \text{if } x < \text{Median} \end{cases} \tag{10}$$

**Case 2: Metric Type = Min (Lower is Better).**

$$S_{norm} = \begin{cases} 1.0 & \text{if } x \leq \text{Gold} \\ 0.5 + 0.5 \cdot \frac{\text{Median} - x}{\text{Median} - \text{Gold}} & \text{if Gold} < x \leq \text{Median} \\ 0.5 \cdot \frac{\text{Median}}{x} & \text{if } x > \text{Median} \end{cases} \tag{11}$$

*Table 7.* Detailed statistics and abbreviation mapping for the 17 selected MLE-Bench tasks. **Task Abbr.**: Abbreviation used in the main text. **Full Task Name**: Original Kaggle competition name. **Metric**: Optimization goal (Max/Min). **Comp.**: Complexity level defined in Section 5.1.1.

| Task Abbr. | Full Task Name | Task Type | Size (GB) | Metric | Comp. |
|---|---|---|---|---|---|
| DISC | detecting-insults-in-social-commentary | Text Classification | 0.00 | Max | Low |
| NYCTaxi | new-york-city-taxi-fare-prediction | Tabular | 5.70 | Min | Low |
| TextNorm | text-normalization-challenge-english-language | Seq2Seq | 0.01 | Max | Low |
| PlantPat | plant-pathology-2020-fgvc7 | Image Classification | 0.80 | Max | Low |
| DocDn | denoising-dirty-documents | Image to Image | 0.06 | Min | Low |
| Birds | mlsp-2013-birds | Audio Classification | 0.59 | Max | Low |
| GQuest | google-quest-challenge | Training LLMs | 0.02 | Max | Medium |
| Arena | lmsys-chatbot-arena | Text Classification | 0.18 | Min | Medium |
| PatentPP | us-patent-phrase-to-phrase-matching | Text Regression | 0.00 | Max | Medium |
| PawPop | petfinder-pawpularity-score | Image Regression | 1.04 | Min | Medium |
| TF-Speech | tensorflow-speech-recognition-challenge | Audio Classification | 3.76 | Max | Medium |
| SaltID | tgs-salt-identification-challenge | Image Segmentation | 0.50 | Max | Medium |
| VentPres | ventilator-pressure-prediction | Forecasting | 0.70 | Min | Medium |
| COVID-Vax | stanford-covid-vaccine | Tabular | 2.68 | Min | High |
| Volcano | predict-volcanic-eruptions-ingv-oe | Signal Processing | 31.25 | Min | High |
| NFL-CD | nfl-player-contact-detection | Video Classification | 5.01 | Max | High |
| MolTrans | bms-molecular-translation | Image to Text | 8.87 | Min | High |

**Process Completion Score** ($S_{comp}$). To quantify the agent's ability to navigate the full end-to-end pipeline-from code generation to valid file submission-we assign a discrete completion score based on the execution state:

- **1.0 (Success)**: Code executes successfully and generates a submission file complying with the required format.

- **0.67 (Format Error)**: Code executes successfully, but the submission file fails format validation.

- **0.33 (Execution Success)**: Code runs without errors, but fails to generate any submission file.

- **0.0 (Failure)**: The pipeline crashes or fails to produce executable code.

# D. Agent System Prompts

To ensure reproducibility, we provide the core system prompts used for the Dynamic Hierarchical Task Network (D-HTN) generation. The planning process is divided into two stages: the **Macro-Planner** (Algorithm 1, Step 3) which generates high-level logical anchors, and the **Micro-Planner** (Algorithm 1, Step 7) which recursively instantiates executable leaf nodes.

## D.1. D-HTN Macro-Planner Prompt

The Macro-Planner is responsible for decomposing the user's global intent into a sequence of high-level parent tasks, utilizing retrieved experience knowledge.

---
**System Prompt: Macro-Planner (Parent Task Generation)**

**Role:** You are a Lead Data Science Architect.
**Context:**

- **Available Task Types:** <Schema Definitions for Data Loading, Preprocessing, Modeling, Evaluation...>

- **Experience Knowledge:** <Retrieved Semantic Priors & Similar Cases>

---

- **User Task Description:** <Original User Query>

**Instruction:** Design a high-level sequence of Parent Tasks to cover the main logic of the implementation flow. You must leverage the provided "Experience Knowledge" to optimize task rationality and order.

For each task, the `instruction` field must briefly describe the goal, input/output scope, and explicitly include:

- Specific file paths involved (if applicable).

- Key constraints (e.g., computational load limits, model complexity constraints).

- Main methods or technical approaches.

**Constraints:** 1. Generate 1 to 5 parent tasks arranged in logical order. 2. Define dependencies in `dependent_task_ids`. 3. Task types must strictly adhere to the "Available Task Types". 4. **Output Format:** Strict JSON list.

**Output Example:**

```
[
  {
    "task_id": "t1",
    "dependent_task_ids": [],
    "instruction": "Load raw data, validate format...",
    "task_type": "Data Loading Layer"
  },
  ...
]
```

## D.2. D-HTN Micro-Planner Prompt

The Micro-Planner evaluates the complexity of each parent task to determine whether it requires further decomposition (Expansion) or can be refined into a single executable instruction (Collapse).

**System Prompt: Micro-Planner (Recursive Instantiation)**

**Role:** You are a Senior Data Scientist AI specialized in converting macro-goals into executable code steps.
**Context:**

- **Current Parent Task:** <Parent Task Instruction>

- **Previous Results:** <Execution Context from Predecessor Nodes>

- **Available Child Task Types:** <Atomic Action Definitions>

**Decomposition Policy (Crucial):** Evaluate the complexity and dependencies of the current task. **Default to NOT splitting (`split=false`)** unless one of the following hard conditions is met:

1. **Runtime Decision Dependency:** Step B depends on the specific runtime output (logs/metrics) of Step A (e.g., determining feature selection strategy after training a baseline).

2. **Long-Running/High-Risk Operations:** Involves heavy training or large downloads requiring checkpoints to prevent total failure.

3. **Extreme Logical Complexity:** Merging steps would severely damage code readability (e.g., complex cleaning + vectorization + clustering).

If the logic is linear and context is sufficient, do not split.

**Output Format (JSON):**

- `split`: boolean

- **If split=true**: Provide a list of `children` tasks (task_id derived like t1.1, t1.2).

- **If split=false**: Provide `refined_parent_instruction` (A precise, code-ready instruction including inputs/outputs/logging requirements based on context).

## D.3. RTR Reconfiguration Prompt

This prompt is activated during the Reactive Topology Reconfiguration phase (Algorithm 1) when an exception signal is intercepted. It acts as the decision core for the self-healing mechanism, determining whether to refine the logic, inject dependencies, or prune the branch.

---

**System Prompt: RTR Recovery Specialist**

**Role:** You are a Data Science Failure Recovery Expert.
**Context:**

- **Failed Task:** <Metadata & Instruction of the crashed node>

- **Error Trace:** <Runtime Exception & Diagnostic Signal>

- **Attempts:** <Current Retry Count (e.g., 2/3)>

- **Pending Tasks:** <Remaining Workflow Topology>

**Strategic Policy:** Analyze the root cause and select one topological action:

1. **REFINE (Retry):** Modify the current task's instruction to fix logic errors or attempt a different approach (e.g., switch to a simpler model, relax constraints, or fix syntax).

2. **SKIP:** If the task is on a non-critical path and skipping it preserves the global goal, bypass it and strictly adjust downstream dependencies to ensure continuity.

3. **INSERT:** Inject a predecessor task to resolve missing dependencies (e.g., insert "Imputation" if "Training" failed due to NaNs).

**Heuristics:**

- **Fallback Principle:** On repeated failures, prioritize simple, robust methods over complex ones to ensure pipeline closure.

- **Integrity:** Ensure the graph remains a DAG (Directed Acyclic Graph) after modification.

**Output Format (JSON):**

```
{
  "action": "REFINE | SKIP | INSERT",
  "reason": "Diagnostic reasoning linking error to action...",
  // Required for REFINE or INSERT:
  "new_task": {
      "task_id": "...",
      "task_type": "...",
```

```
        "instruction": "Refined or New Instruction...",
        "dependent_task_ids": [...]
    },
    // Required for SKIP:
    "adjusted_pending_tasks": [
        { "task_id": "...", "dependent_task_ids": [...] },
        ...
    ]
}
```

## D.4. Semantic Reflection & Experience Distillation Prompts

This section details the prompts used for the "Dual-loop" mechanism: the **Verifier** and **State Projector** (Inner Loop) ensure immediate execution consistency, while the **Distiller** and **Archivist** (Outer Loop) compress trajectories into reusable long-term knowledge.

---

**System Prompt: Execution Verifier (Semantic Reflection)**

**Role:** You are a Strict Execution Monitor. Determine if the task was truly successful based on the execution logs.
**Input:**

- **Instruction:** <Original Micro-Operator Command>

- **Result:** <Stdout/Stderr Logs>

**Review Criteria (Strict):** Only check for explicit runtime errors captured and printed (e.g., keywords like "Error:", "Exception:", "Traceback").

- **FAIL:** If such explicit error keywords appear in the logs.

- **SUCCESS:** If no explicit error keywords are found (even if logic seems incomplete).

- **NOTE:** Do not speculate on potential logical bugs; focus solely on caught exceptions.

**Output (JSON):** `{ "success": true/false, "reason": "..." }`

---

**System Prompt: Inner-Loop State Projector**

**Role:** You are a Data Science State Analyst. Analyze the execution result to extract metadata for the next step.
**Tasks:**

1. **Extract New Knowledge:** Return key-value pairs of persistent artifacts. Focus on:
   - **File Paths:** Absolute/relative paths of generated .csv, .pkl, .png files.
   - **Variables:** Key DataFrame names (df_train), Model objects.
   - **Meta-Info:** target_col, id_col, data shapes.
   - **Model Init Code (Crucial):** The exact code used to initialize the model (e.g., `RandomForest(n_est=100)`) for reconstruction.

2. **Suggest Next Steps:** Based on the result (e.g., if accuracy is low → tune; if data has NaNs → clean).

**Output (JSON):**

`{`

---

```
   "new_knowledge": { "cleaned_path": "...", "model_init": "..." },
   "suggested_next_steps": ["Step 1...", "Step 2..."]
}
```

### System Prompt: Knowledge Distiller (Noise Filtration)

**Role:** You are a Senior Research Fellow conducting a "Post-Mortem" analysis.
**Objective:** Extract **High-Value, Novel, or Counter-Intuitive** insights from this task. STRICTLY FILTER out common knowledge (e.g., "Use GridSearch", "Handle NaNs").

**Allowed Distillation Categories:**

- **Architectural Advice:** Non-standard pipelines or corner-case adjustments.

- **Feature Engineering:** Domain-specific hard-core features or encoding improvements.

- **Modeling:** Custom Loss functions, specific layer connections, or "Anti-Common Sense" hyperparameter settings.

**Output Schema (JSON Array):** Return `[]` if no novel insight is found. Otherwise:

- `condition`: Triggering condition (e.g., "sparsity > 98%").

- `recommendation`: The specific innovation.

- `anti_pattern`: Common practices that failed in this specific context.

- `rationale`: Empirical evidence of improvement.

### System Prompt: Case Archivist (Outer-Loop Memory)

**Role:** You are a Knowledge Base Curator. Summarize the entire successful episode into a teaching case.
**Goal:** Create a structured entry for the Expert Library ($\mathbb{G}$) to assist future agents.

**Output Fields (JSON):**

1. `title`: Concise problem summary (e.g., "Memory Optimization for High-Cardinality Features").

2. `description`: Task goal, data characteristics, and main challenges.

3. `solutions`: Dictionary of key techniques used (keys: "Feature Strategy", "Model Selection", etc.).

4. `tags`: 3-5 keywords (Algorithm, Data Type, Specific Trick).

## E. Supplementary Evaluation Metrics and Sensitivity Analysis

To ensure transparency and provide a comprehensive view of the agents' capabilities beyond our custom scoring metric, we supplement our main evaluation with standard competition metrics (medal tiers) and a sensitivity analysis of our comprehensive score weighting.

### E.1. Medal-Tier and Task Completion Summary

While the normalized score ($S_{norm}$) effectively standardizes performance across diverse metrics, reporting standard Kaggle-style medal tiers provides an intuitive benchmark of absolute prediction quality. Table 8 summarizes the medal rates, the absolute number of medaled tasks, and the number of successfully completed tasks across the evaluated frameworks.

As shown in Table 8, $R^3$DAO achieves the highest task completion rate (successfully executing 16 out of 17 tasks) while securing the second-highest medal rate (29.5%). Although the high-resource R&D-Agent achieves a higher medal rate, it

*Table 8.* Medal-tier summary and task completion across different frameworks on the 17-task benchmark. $R^3$DAO demonstrates the highest execution robustness.

| Model | Medal Rate (%) | Medal Tasks | Successful Tasks |
|---|---|---|---|
| R&D-Agent | 41.2 | 7 | 9 |
| $R^3$**DAO (Ours)** | **29.5** | **5** | **16** |
| AIDE | 11.8 | 2 | 9 |
| OpenHands | 5.9 | 1 | 10 |
| DataInterpreter | 5.9 | 1 | 4 |
| MLAB | 0.0 | 0 | 8 |

suffers from a significantly lower completion rate (completing only 9 out of 17 tasks). This demonstrates that $R^3$DAO offers the best balance between execution reliability and competitive prediction performance among all evaluated systems.

### E.2. Sensitivity Analysis of the Comprehensive Score

In Section 5.1.2, we defined the comprehensive score as $S_{total} = 0.6 \times S_{norm} + 0.4 \times S_{comp}$ to concurrently evaluate prediction accuracy and execution robustness. To ensure that our findings are robust to this specific hyperparameter choice, we present a sensitivity analysis across varying weight combinations in Table 9.

*Table 9.* Sensitivity analysis of the comprehensive score under varying weights for normalized performance ($S_{norm}$) and completion score ($S_{comp}$).

| Weighting ($S_{norm}$ / $S_{comp}$) | R&D-Agent | $R^3$**DAO (Ours)** | AIDE |
|---|---|---|---|
| 1.0 / 0.0 | **0.444** | 0.422 | 0.263 |
| 0.75 / 0.25 | 0.465 | **0.552** | 0.330 |
| 0.5 / 0.5 | 0.487 | **0.682** | 0.396 |
| 0.25 / 0.75 | 0.508 | **0.811** | 0.463 |
| 0.0 / 1.0 | 0.529 | **0.941** | 0.529 |

The results indicate that $R^3$DAO's relative advantage grows as the weight of the completion rate ($S_{comp}$) increases. This aligns with the fundamental intuition of autonomous data science orchestration: a pipeline that crashes or fails to produce a valid submission inherently yields zero prediction quality. Notably, even under the extreme setting where only raw performance is considered ($1.0/0.0$), $R^3$DAO remains highly competitive (0.422 vs. R&D-Agent's 0.444).

Regardless of the chosen weighting scheme, $R^3$DAO's 94.1% overall success rate and its minimal resource footprint (103k tokens per task) remain its most distinguishing features. As supported by the data, no other baseline achieves a success rate above 59% or exhibits comparable token efficiency.

