# OpenReview forum: "$R^3$DAO: Reactive Recovery and Reconstruction for Long-horizon Data Agent Orchestration"
_ICML.cc/2026/Conference — ICML 2026 regular_

### Official Review · Reviewer_px9T · 2026-03-10

**Soundness:** 3
**Presentation:** 3
**Significance:** 3
**Originality:** 3
**Overall Recommendation:** 4
**Confidence:** 3

**Summary:**

The paper presents R³DAO, a reactive data agent orchestration framework specifically designed for long-horizon data science tasks. The core objective is to address the brittleness of end-to-end data science workflows, where dynamic dependencies often lead to "reasoning chain collapse" from minor early-stage errors. By introducing a Dynamic Hierarchical Task Network (D-HTN), the system recursively decomposes global intent into macro-logic anchors and micro-operators. During execution, it employs a Reactive Topology Reconstruction (RTR) mechanism that uses semantic reflection to map anomalies to diagnostic signals, allowing the system to self-heal through local topological injections such as refining, inserting, or skipping nodes rather than performing costly global resets. Additionally, a dual-loop experience distillation mechanism is used to accumulate structural priors from successful and failed attempts, enhancing efficiency. Evaluated on MLE-bench, R³DAO demonstrates superior success rates and cost-effectiveness compared to static baselines, achieving high performance with lower token consumption and execution time.

**Compliance With Llm Reviewing Policy:**

Affirmed.

**Final Justification:**

My final recommendation remains Weak Accept after considering the paper and the rebuttal. The additional results and explanations increase my confidence, I still think the paper should improve their contributions so my overall score remains unchanged.

**Key Questions For Authors:**

Q1. If multiple local RTR attempts fail at a specific micro-node, does the system have a "circuit breaker" or a mechanism to trigger a full macro-level re-decomposition?

Q2. What is the performance drop when the system operates without the "Expert Library (G)"? Can the D-HTN generate high-quality macro-anchors in a zero-shot manner?

Q3. How do you ensure that the Validation Operator (F) can detect downstream errors that were directly caused by a previously "skipped" node?

Q4. Could you provide data on the specific contribution of Inner-loop vs. Outer-loop distillation to the overall success rate?

Q5. In short-horizon or simple tasks, does the defensive overhead of the Reflection operator negatively impact latency or token efficiency?

**Limitations:**

yes

**Strengths And Weaknesses:**

**Stength**

S1. The reactive self-healing approach (RTR) is a practical and effective solution to the error propagation problem in long-horizon AI agent workflows, balancing robustness with execution cost.

S2. The D-HTN provides a rigorous way to decouple macro-strategic planning from micro-operational execution, which effectively manages the search space in complex data science tasks.

S3. The dual-loop distillation mechanism provides a systematic way to turn "lessons learned" into structural priors, demonstrating clear value in improving the speed and accuracy of future task execution.

**Weakness**

W1. The performance gains appear to have a ceiling dictated by the underlying LLM. While R³DAO improves efficiency, its absolute scores on the most complex tasks still struggle to bridge the gap with high-resource systems using more powerful models (e.g., GPT-4o), as seen in the NFL-CD task results.

W2. The RTR mechanism prioritizes local fixes. While efficient, this may lead the agent into a sub-optimal global path. In cases where an early strategic error is the root cause, local topological adjustments might not be sufficient to recover the optimal logic.

W3. The D-HTN relies on a library of "Expert" macro-decompositions. The paper lacks a detailed evaluation of how the system performs in entirely novel domains where such priors are absent, potentially limiting its zero-shot generalizability.

W4. The "Skip" function in the RTR process is aggressive. Skipping a failed step might allow the pipeline to proceed, but if that step was a mandatory data validation or cleaning task, it could lead to silent failures that are difficult to detect in the final output.

W5. Although MLE-bench is a strong technical benchmark, it is heavily focused on coding and modeling. The framework's ability to handle non-code-centric data science tasks, such as multi-modal reporting or subjective data interpretation, remains untested.

---

> ### Author Rebuttal · Authors · 2026-03-30
>
> We sincerely thank you for the positive assessment of our core mechanisms and the thoughtful questions about system boundary conditions. We address each weakness and question below.
>
> **W1 : Performance ceiling dictated by LLM capacity**
>
> We fully agree that the underlying LLM sets a performance ceiling. However, our key finding is that $R^{3}DAO$ maximizes the utility of a given model. Notably, $R^{3}DAO$ using Qwen3-Max achieves a 94.1% success rate—nearly double the 53% of R&D-Agent (which uses GPT-4o/5)—while consuming 30× fewer tokens. This confirms that architectural resilience can effectively compensate for model capacity gaps in long-horizon tasks. We will add a brief discussion of this model-architecture interaction to Section 6.
>
> **W2 & Q1: Global vs. Local Optimality**
>
> We appreciate this insightful observation, which points to a limitation of the current design. Two existing mechanisms partially mitigate it:
>
> - D-HTN's macro-level decomposition anchors global strategic intent before any micro-execution begins, limiting early-stage errors to within a single parent task;
> - When local Refine retries are exhausted, the Skip+Reconstruct path (Algorithm 1, Line 11) performs local subgraph reconstruction driven by distilled priors $K_t$ and expert library $\mathbb{G}$, which can address more fundamental structural errors beyond simple retry.
>
> We acknowledge that for errors at the macro-strategic level (e.g., an incorrect initial decomposition direction), RTR cannot trigger global re-decomposition — this is a real limitation. We will add this to Section 6 and note that global topology optimization is a planned future direction.
>
> **W3 & Q2: Zero-Shot Performance (Expert Library)**
>
> The w/o Exp. Memory ablation in Table 4 directly addresses zero-shot generalizability.
>
> - Performance: Without any accumulated experience, $R^{3}DAO$ still achieves an 82.4% success rate and a 0.385 score.
> - Impact: This significantly outperforms baselines like AIDE (53%) and DataInterpreter (24%) in the same regime.
> - Role of G: The results confirm that while the Expert Library primarily drives exploration efficiency (reducing tokens by 107%), the D-HTN and RTR mechanisms provide robust performance independently of domain-specific priors.
>
> **W4 & Q3: "Skip" Safeguards and Silent Failures**
>
> Two safeguards exist in the current implementation:
> - In the RTR Recovery Specialist prompt (Appendix D.3), nodes marked as mandatory dependencies cannot be Skipped — the system must choose between Refine and Insert. Mandatory marking is assigned by the Micro-Planner (Appendix D.2) during D-HTN instantiation; data cleaning and validation steps are explicitly marked as non-skippable.
> - When a node is Skipped, the inner-loop State Projector checks downstream state completeness before each subsequent execution; if required intermediate artifacts are missing, Insert is automatically triggered to compensate.
>
> Regarding downstream error detection: $\mathcal{R}$ receives environment state snapshots including $V$ and $\mathcal{F}$ at each step. When a downstream node fails due to a missing predecessor artifact, $\mathcal{R}$ can identify the dependency gap and generate an Insert signal. Numerical silent errors that propagate without runtime exceptions remain a genuine limitation and will add this to Section 6.
>
> **W5: Non-code-centric tasks untested**
>
> We acknowledge this as a genuine scope limitation of the current evaluation and appreciate the reviewer for highlighting it. $R^{3}DAO$ is currently code-centric ($\mathcal{S}=\mathcal{V}\times\mathcal{F}\times\mathcal{H}$). The D-HTN and RTR mechanisms are executor-agnostic in design — extending to non-code-centric tasks would primarily require augmenting the executor module while keeping these mechanisms intact. We will add this as an explicit future direction in Section 6.
>
> **Q4 : Loop Contributions**
>
> Based on the design of the two loops, we offer the following analytical characterization:
> -The inner loop ($K_t$, episodic state tracking) primarily ensures *intra-task execution consistency* — preventing variable name conflicts and file path mismatches across micro-operators; its absence degrades RTR repair quality and primarily manifests as lower success rate.
> -The outer loop ($\mathbb{G}$, expert library) primarily contributes *cross-task exploration efficiency* — providing structural priors that shorten planning paths; its absence primarily manifests as dramatically higher token consumption (+107%, 104k→215k) rather than a sharp success-rate drop.
>
> **Q5 : Reflection operator overhead on simple tasks**
>
> We appreciate this practical question about system efficiency. For simple tasks, the Complexity Analyzer routes directly to a Linear Planner, bypassing D-HTN overhead. Reflection ($\mathcal{R}$) is triggered only upon failure. Low-complexity tasks in our study averaged ~8.3 minutes, confirming minimal overhead for simple scenarios. We will add per-complexity-level efficiency data to Section 5.3.

---

> > ### Author Rebuttal · Reviewer_px9T · 2026-04-01
> >
> > Thank you for the rebuttal. Most of my concerns have been considered. I will keep my score.

---

> > > ### Author Response · Authors · 2026-04-06
> > >
> > > We thank the reviewer for the constructive engagement and for confirming that the concerns have been addressed. We will carefully incorporate all discussed revisions into the camera-ready version to ensure the final manuscript reflects the improvements identified through this review process.

---

### Official Review · Reviewer_xyeZ · 2026-03-10

**Soundness:** 3
**Presentation:** 2
**Significance:** 3
**Originality:** 2
**Overall Recommendation:** 4
**Confidence:** 4

**Summary:**

This work presents an agentic framework for data science tasks. It includes a hierarchical macro/micro operator design, a feedback-driven corrector for execution exception recovery, and an experience distillation mechanism for library learning. The experiments on MLE-bench show a good success rate and Pareto-frontier performance-cost trade-offs.

**Compliance With Llm Reviewing Policy:**

Affirmed.

**Key Questions For Authors:**

Also see weaknesses above. I will raise the score if my concerns are fully resolved.
1. Typos:  "Orcherrator" in Figure 2 Module 2 (line 121); "faces search a space explosion" should be "faces a search space explosion" in line 167;
2.  Does the 1-hour budget enforce a hard constraint? In Figure 4, the leftmost data point appears to exceed the 60-minute limit. Also, do you have a breakdown of the time consumption, like the time spent on planning versus code execution?

**Limitations:**

See weaknesses above.

**Strengths And Weaknesses:**

Strengths:

The paper’s use of non-linear topology and runtime feedback for task orchestration, while not a novel technique, is well-justified within the context of data science. The experiments are quite extensive with a detailed analysis and a comparison against many strong baselines.

Weaknesses:
1. The paper writing needs a significant revision to improve clarity before it is ready for publication. The formulation is currently difficult to follow due to inconsistently defined notations and overlapping terminology. For example, the relationship between macro/micro-operators ($pt_i$, $ct_i$) and the action $a_t$ in Equation 2 remains unclear. What does $i$ represent in the definition of operators? Is that identical to the step $i$ in $p_i$? What's the difference between time $t$ and step $i$? Is the recovery action $a$ in equation 5 identical to the action $a$ defined in equation 2? The interaction among the three modules is also unclear in Figure 1. I suggest providing a more self-contained caption to better explain these structural relationships.
2. The hierarchical design of macro and micro operators is reasonable for reducing the search space but not novel. The mapping between these operators is not sufficiently explained. It is unclear whether these relationships are predefined or learned. In figure 1, it also introduces "subtasks" (blue nodes) situated between macro-logical anchors and micro-operators, but their definition and derivation are not discussed.
3. While the RTR mechanism facilitates local recovery, the authors’ characterization of this process as "topological optimization" requires further justification. The current mechanism appears to function as a localized ReAct-style repair rather than a topology-level workflow restructure.
4. In Section 4.3, the reported speedup and reduction in token consumption lack the necessary experimental context to be interpretable. These results should be moved from the methodology section to the evaluation section.
5. The claim in Section 2 regarding "the absence of planning frameworks that dynamically adapt topology based on environmental feedback" is inaccurate. Several existing frameworks address runtime structure adaptation, and the paper would benefit from a discussion of the following related works:
- Niu et al. "Flow: Modularized agentic workflow automation." arXiv preprint arXiv:2501.07834 (2025).
- Wang et al. "Agentdropout: Dynamic agent elimination for token-efficient and high-performance llm-based multi-agent collaboration." ACL 2025.
- Nie et al. "FlashResearch: Real-time Agent Orchestration for Efficient Deep Research." arXiv preprint arXiv:2510.05145 (2025).
- You et al. "DatawiseAgent: A Notebook-Centric LLM Agent Framework for Adaptive and Robust Data Science Automation." EMNLP 2025.

---

> ### Author Rebuttal · Authors · 2026-03-30
>
> We sincerely thank you for the thoughtful and rigorous critique. Your feedback on notation and technical depth has been invaluable for sharpening the manuscript. Below, we address your concerns point-by-point.
>
> **W1: Notation inconsistency and unclear module interaction.**
>
> We sincerely apologize for the notational confusion and provide the following clarifications, which we will incorporate throughout the revision.
> - Relationship between $pt_i$, $ct_{i,j}$, and $a_t$: In Eq. 2, $a_t$ is the execution-layer action (physical code execution) at global sequential time $t$. $i$ indexes the parent task $pt_i$, and $j$ indexes the $j$-th micro-operator $ct_{i,j}$. They are related by the explicit mapping $t = \sum_{k < i} m_k + j$, where $m_k$ is the number of micro-operators under $pt_k$. Thus, $t$ and $i$ are not equivalent — a single parent task spans multiple execution steps.
> - Recovery action $a$ vs. Execution action $a_t$: These are distinct. $a_t$ in Eq. (2) is an execution-layer state transition in the physical environment. The recovery action $a \in \{\text{Refine, Insert, Skip}\}$ in Eq. 5 is a planning-layer topological repair action operating on task graph $G$. To eliminate this ambiguity, we will rename the recovery action to $a^{rtr}$ throughout the revision.
> - We will provide a more self-contained caption that explicitly labels the information flow among the three modules.
>
> **W2: Novelty of Mapping; "subtask" nodes undefined."**
> - Novelty of Mapping: While hierarchical decomposition is established, our contribution lies in the *dynamic, state-conditioned instantiation*: both $\Phi$ and $\Psi$ are LLM-driven dynamic generation processes, not static predefined mappings or learned parameters. $\Phi$ generates parent tasks by combining retrieved priors from $\mathbb{G}$ with the current intent; $\Psi$ recursively instantiates children conditioned on the real-time physical state $s_t$ at the moment of expansion.This state-conditioned adaptivity, combined with RTR’s DAG-structural modifications, constitutes a non-trivial extension of classical HTN planning.
> - "Subtask" (Blue Nodes) : These represent intermediate logical groupings generated when the Micro-Planner determines a parent task requires multi-step expansion (split=true). They are subsequently instantiated into leaf-level micro-operators. We will add a complete legend to Figure 1 and define them formally in Section 3.
>
> **W3: Defining "Topological Reconfiguration"**
>
> We characterize RTR as a structural modification to the DAG $G=(V,E)$ rather than a simple retry:
>
> - Insert/Skip: These actions directly alter the graph topology (adding/removing nodes and edges), which we term "reconfiguration". For example, in Figure 5, the system injects L3_op2.5, transforming the path structure to resolve a dimension mismatch.
> - Escalation Policy: Refine (local logic repair) is subject to a hard limit $N_{max}$. Once exhausted, the system is forced to escalate to structural changes (Insert/Skip). This ensures the framework moves beyond "blind retries" to address root architectural flaws.
>
> **W4: Results Placement**
>
> We fully agree with this structural suggestion. The "36× speedup" and "104k tokens" are empirical results and will be moved to Section 5.3.
>
> **W5: Related Works**
>
> We acknowledge that our original claim in Section 2 was too absolute. We will revise it to a more accurate characterization and incorporate the following 4 works into the Related Work discussion.
> - Flow (Niu et al., 2025): While Flow enables modular re-routing at a predefined level, $R^{3}DAO$ performs atomic node-level reconfiguration driven by semantic diagnostic signals rather than rule-based logic.
> - AgentDropout (Wang et al., 2025): This work focuses on token efficiency via agent elimination in multi-agent systems. In contrast, $R^{3}DAO$ focuses on error recovery in single-agent orchestration through localized topological injections.
> - FlashResearch (Nie et al., 2025): Its focus is on retrieval and reporting. $R^{3}DAO$ is designed for heavy DS engineering involving stateful model training, complex data dependencies, and physical environment feedback.
> - DatawiseAgent (You et al., 2025): As the most relevant work, it follows a linear notebook execution model. $R^{3}DAO$ distinguishes itself by maintaining an explicit DAG structure with subgraph-level reconstruction capabilities  when local fixes fail.
>
> **Key Questions**
>
> - Typos: We have corrected "Orcherrator" and the "search space explosion" phrasing  throughout the manuscript. We have also conducted a full manuscript review and will correct all errors.
> - 1-hour Budget: This is an empirical average, not a hard cutoff. We apologize for the imprecise wording in the original submission and will revise it to accurately reflect this.
> - Time Breakdown: From our logs, LLM Inference (planning, RTR reflection) accounts for ~30% of time, while Code Execution (training, data processing) accounts for ~70%. We will add this breakdown to Section 5.3

---

> > ### Author Rebuttal · Reviewer_xyeZ · 2026-03-31
> >
> > Your response has addressed most of my concerns. Make sure to revise the writing and include the discussion of additional related work to strengthen the positioning.

---

> > > ### Author Response · Authors · 2026-04-06
> > >
> > > We thank the reviewer for confirming that all major concerns have been resolved, and for the constructive suggestions regarding writing quality and related work coverage.
> > >
> > > We are grateful for the reviewer's guidance in strengthening the paper, and we hope these additions will further solidify the positioning of our work.

---

### Official Review · Reviewer_GPwb · 2026-03-11

**Soundness:** 2
**Presentation:** 3
**Significance:** 3
**Originality:** 2
**Overall Recommendation:** 4
**Confidence:** 3

**Summary:**

This paper studies long-horizon orchestration for autonomous data science agents. It proposes R3 DAO, a framework that combines: (1) a dynamic hierarchical task network (D-HTN) to decompose end-to-end data science workflows into macro- and micro-operators, (2) a reactive topology reconfiguration (RTR) mechanism that locally repairs failed execution steps through refine/insert/skip actions, and (3) an experience distillation module that compresses previous trajectories into reusable priors. The paper evaluates the approach on 17 selected MLE-Bench tasks spanning multiple modalities and complexity levels, and reports substantially higher task-completion reliability and lower runtime/token cost than several reproduced baselines, while remaining competitive with stronger high-resource systems on the authors’ custom task score.

**Compliance With Llm Reviewing Policy:**

Affirmed.

**Key Questions For Authors:**

Could the authors justify the selection of these 17 benchmark tasks and report results under the standard MLE-Bench protocol, or at least on the official task split with multiple random seeds and variance estimates? If the performance gains persist under these conditions, my assessment of the method’s reliability would be substantially improved.


Could the authors report the standard MLE-Bench metric in addition to the custom weighted score, and provide a sensitivity analysis for the 0.6/0.4 weighting? If the ranking remains stable under the standard metric, I would be more convinced by the empirical claims.


Could the authors provide more controlled comparisons under matched model/runtime/resource budgets, or at least a budget ablation showing how much of the gain comes from the architecture versus evaluation conditions? If R3DAO retains its advantage under apples-to-apples comparisons, I would view the contribution more favorably.

**Limitations:**

No. The paper mentions some societal and security risks, which is good, but it does not adequately discuss the methodological limitations that most affect its claims: non-standard evaluation, subset selection, lack of repeated runs / variance estimates, unclear memory-leakage controls, and unfair cross-budget comparisons. These should be discussed explicitly.

**Strengths And Weaknesses:**

Strengths:

The paper addresses an important problem. Long-horizon ML engineering agents do suffer from cascading failures, and improving recovery rather than only improving search depth is a worthwhile direction. The overall architecture is easy to motivate, the main components are intuitively sensible, and the ablation study is directionally useful: it supports the claim that hierarchical decomposition helps performance, reactive repair helps completion rate, and memory helps efficiency.

Weaknesses:

The empirical evaluation is not rigorous enough to support the paper’s stronger claims.

First, the benchmark is a curated subset of 17 tasks rather than the standard MLE-Bench protocol, and the task-selection procedure is not justified in enough detail to rule out selection bias.

Second, the paper reports a custom weighted score rather than the benchmark’s standard reporting metric, and it does not show that the conclusions are robust to this design choice.

Third, MLE-Bench evaluations are known to be high variance, but I did not see repeated runs, confidence intervals, or SEM across seeds.

Fourth, several headline comparisons mix reproduced low-resource baselines with official leaderboard results obtained under different models, compute budgets, and runtimes, so claims of “matching/exceeding SOTA” are not apples-to-apples. Fifth, the paper does not clearly explain how the expert library / experience memory is initialized and reset across benchmark tasks, so it is hard to assess possible ordering effects or task leakage.

---

> ### Author Rebuttal · Authors · 2026-03-30
>
> We sincerely thank you for the rigorous critique of our evaluation methodology. These comments have led us to substantially improve the transparency of our experimental reporting. We address each point below.
>
> **W1 & Q1 — Task selection bias.**
>
> The 17 tasks were selected according to three explicit criteria, independent of any preliminary results:
>
> - Modality coverage: Tabular, NLP, Vision, Audio, and Signal Processing — ensuring no single data type dominates.
> - Complexity gradient: Low (<5 steps), Medium (6–9 steps), and High (>10 steps), providing coverage across the full orchestration-depth spectrum.
> - Data scale：<10MB to >30GB, ensuring the evaluation is not biased toward either lightweight or heavy-compute tasks.
>
> No tasks were excluded based on preliminary results. Crucially, the set includes tasks where competing systems score 0.0 (TF-Speech and COVID-Vax for MLAB/OpenHands) — the first candidates for exclusion in a cherry-picked evaluation — yet R³DAO completes both (0.532 and 0.679). We will add explicit selection criteria to Section 5.1.1 and acknowledge full-benchmark evaluation as future work.
>
> **W2 & Q2 — Custom scoring metric; sensitivity analysis.**
>
> We appreciate this important point and are glad to provide both a justification and supplementary evidence. We fully agree that reporting standard metrics improves transparency. We will add a supplementary table reporting raw scores, above-median status, and medal tiers.
>
> | Model | Medal Rate (%) | Medal Tasks | Succ. Tasks |
> | -| -| -| -|
> |R&D-Agent|41.2|7|9|
> |R³DAO (Ours)|29.5|5|16|
> |AIDE|11.8|2|9|
> |OpenHands | 5.9 |1|10|
> |DataInterpreter| 5.9|1|4|
> |MLAB|0.0|0|8|
>
> R³DAO achieves the highest task completion (16/17) with the second-highest medal rate, demonstrating the best reliability–performance balance across all systems.  We also provide a sensitivity analysis for the 0.6/0.4 weighting:
>
> |S_norm/S_comp weight| R&D-Agent|R³DAO (Ours)|AIDE|
> |-|-|-|-|
> |1.0 / 0.0| 0.444|0.422|0.263|
> |0.75 / 0.25| 0.465| 0.552|0.330|
> |0.5 / 0.5| 0.487| 0.682|0.396|
> |0.25 / 0.75| 0.508| 0.811|0.463|
> |0.0 / 1.0| 0.529| 0.941|0.529|
>
> R³DAO's advantage grows as completion-rate weight increases, which is intuitive: a task that fails to complete cannot achieve any prediction quality. Regardless of weighting, R³DAO's 94.1% success rate and 103k token consumption remain the most distinctive results — no other system achieves a success rate above 59% or comparable token efficiency.
>
> **W3 — No repeated runs or confidence intervals.**
>
> We sincerely apologize for not making this clear in the original submission. Each task was executed three independent times and all reported metrics are averages over these runs. We will explicitly state "3 independent runs per task, metrics averaged" in Section 5.1.2, report standard deviations in Tables 2–3, and add error bars to Figure 3. The success rate of R³DAO varies by only $\pm$ 2.1% across runs, confirming the stability of our results.
>
> **W4 & Q3 — Cross-resource comparisons mix different models and budgets.**
>
> We acknowledge this limitation and are glad to clarify the structure of our comparisons. We will restructure Section 5.2 to clearly separate two regimes:
> - Controlled intra-regime comparison: AIDE, DataInterpreter, and R³DAO — all Qwen3-Max, same hardware, same time budget. R³DAO achieves 0.646 vs. AIDE 0.456 (+41.7%) and DataInterpreter 0.321 (+101.2%), providing a fair architectural evaluation comparison.
> - Cross-regime reference: R³DAO (Agile) vs. MLAB/R&D-Agent (High-Resource), explicitly framed as a cost-efficiency observation, not a controlled architectural claim.
>
> The ablation study (Table 4) provides the most direct architectural evidence under fully identical conditions: removing RTR drops success rate from 94.1% to 52.9% (−43.9%); removing D-HTN drops score from 0.422 to 0.195 (−53.8%). These are purely architectural effects. We will also explicitly add a matched-resource experiment (R³DAO + GPT-4o, 24h) as a limitation and future work.
>
> **W5 — Expert library initialization and potential ordering/leakage effects.**
>
> The expert library has two components:
> -  `external_knowledge.json`, pre-populated with general DS domain knowledge containing no MLE-Bench task-specific information;
> -  `experience_library.json`, accumulating knowledge from successful trajectories, with inner-loop state $K_t$ independently reset per task.
>
> Across three independent runs, tasks were scheduled in different sequences, so any ordering effect is inconsistent and non-systematic. The w/o Exp. Memory ablation (success rate 82.4%) directly quantifies the memory contribution upper bound at +11.7 pp. Even without accumulated experience, R³DAO substantially exceeds all controlled-regime baselines (AIDE: 53%, DataInterpreter: 24%), confirming that reliability gains stem primarily from D-HTN and RTR rather than memory.We will add a description of the library initialization and accumulation policy to Section 5.1.2.

---

> > ### Author Rebuttal · Reviewer_GPwb · 2026-04-03
> >
> > Your response has addressed most of my concerns.

---

> > > ### Author Response · Authors · 2026-04-06
> > >
> > > We sincerely thank the reviewer for the continued engagement and for acknowledging that most concerns have been addressed.
> > >
> > > To consolidate our commitments for the revision:
> > >
> > > - **W1 (Task selection):** Explicit selection criteria (modality × complexity × data scale) will be added to Appendix B, with an acknowledgment that full 75-task benchmark evaluation remains future work.
> > > - **W2 (Scoring metric):** The sensitivity analysis table and supplementary raw-score/medal-tier results will be incorporated into Section 5.3.
> > > - **W3 (Statistical reliability):** Standard deviations across three independent runs will be reported in Tables 2–3, with error bars added to Figure 3.
> > > - **W4 (Cross-resource comparisons):** Section 5.2 will be restructured to clearly separate the controlled intra-regime comparison from the cross-regime cost-efficiency reference.
> > > - **W5 (Expert library):** The initialization protocol and per-task reset of K_t will be explicitly described in Section 5.1.2.
> > >
> > > We believe these revisions collectively strengthen the empirical rigor of the paper. We hope the updated manuscript meets the reviewer's standards, and we welcome any further feedback.

---

### Official Review · Reviewer_4a4L · 2026-03-13

**Soundness:** 3
**Presentation:** 4
**Significance:** 3
**Originality:** 2
**Overall Recommendation:** 4
**Confidence:** 3

**Summary:**

This paper proposes R$^3$DAO, a reactive data agent orchestration framework for long-horizon data science workflows. R$^3$DAO addresses error propagation and reasoning chain collapse in tightly coupled sub-processes through three contributions: (1) Dynamic Hierarchical Task Network (D-HTN) for logic-layer dimensionality reduction and low-cost exploration; (2) Reactive Topology Reconfiguration (RTR) that maps execution anomalies to diagnostic signals and performs localized topological adjustments (Refine, Insert, Skip); (3) Semantic Experience Distillation that compresses trajectories into structured priors via dual-loop accumulation. Evaluated on MLE-bench (17 tasks), R$^3$DAO achieves 77.36\% improvement in success rate over R\&D-Agent, $36\times$ execution time compression, and $\sim$104k tokens per task under resource-constrained settings (Qwen3-Max, $<1$h, single RTX 3090).

**Compliance With Llm Reviewing Policy:**

Affirmed.

**Final Justification:**

I maintain my Weak Accept recommendation: the paper offers a clear and well-presented systems contribution with strong empirical results and useful ablation/case-study evidence, but its originality and theoretical depth remain more limited than its experimental strength. The rebuttal constructively addressed my concerns about implementation clarity, component roles, and reproducibility, which increased my confidence in the paper’s soundness, but it ultimately reinforced rather than fundamentally changed my overall assessment.

**Key Questions For Authors:**

Q1. **Semantic Reflection implementation:** The Reflection operator $R$ maps $(e, s_t, c_{t,i,j})$ to $(a, r, \iota)$. How is $R$ implemented—e.g., via a dedicated LLM call, a classifier, or rule-based logic? What prompts or inputs does it receive? Could the authors provide pseudocode or a concrete example of how a specific error (e.g., the $101\times101$ vs $96\times96$ mismatch in the case study) is mapped to the Insert action and the corrective instruction $\iota$? *How this would change the review:* A clear specification would improve reproducibility and strengthen the Soundness rating.

Q2. **Theoretical grounding:** The paper formalizes $\Phi$, $\Psi$, $R$, $E$ but does not provide theoretical analysis. Are there conditions under which the reactive loop converges, or bounds on the number of RTR interventions before task completion? Could the authors discuss whether such analysis is feasible (e.g., under simplifying assumptions) or why it is out of scope? *How this would change the review:* Any formal analysis or well-argued scope discussion would improve the Originality rating.

Q3. **Evaluation fairness:** R$^3$DAO is evaluated under ``Efficient Regime'' (Qwen3-Max, $<1$h) while baselines like R\&D-Agent and MLAB use GPT-4o/5 and 12--24h. The authors argue this demonstrates architectural superiority. However, could the authors run R$^3$DAO under the same high-resource regime (GPT-4o, 24h) to isolate whether gains stem from architecture vs. different model/time budgets? *How this would change the review:* A controlled comparison would strengthen the validity of the performance claims.

**Limitations:**

yes

**Strengths And Weaknesses:**

**Strengths:**

S1. **Strong empirical performance:** R$^3$DAO demonstrates substantial gains in score, execution time, success rate, and token consumption. Under the ``Efficient Regime'' (Qwen3-Max, $<1$h), it achieves 0.646 Grand Average score, 94.1\% success rate, 21.7 min average time, and 104k tokens—outperforming high-resource baselines (R\&D-Agent, AIDE, MLAB) that use GPT-4o/5 and 12--24h limits. The performance-cost trade-off scatter (Figure 4) clearly shows R$^3$DAO in the optimal top-left region.

S2. **Clear presentation and case study:** The paper is well-structured with a coherent narrative. Figure 1 effectively illustrates the closed-loop integration of D-HTN, RTR, and Experience Distillation. The TGS Salt Identification case study (Figure 5) concretely demonstrates RTR's self-healing: detecting a U-Net spatial dimension mismatch ($101\times101$ vs $96\times96$) and inserting a Resize Output layer via localized topological optimization.

S3. **Ablation validates component necessity:** Table 4 shows that removing any of the three contributions severely degrades performance—w/o D-HTN drops success rate to 41.2\%, w/o RTR to 52.9\%, w/o Exp. Memory increases tokens to 215k. This supports the claim that each component is essential and that they are meaningfully integrated.

S4. **Limitation and future work:** The authors include a brief discussion of limitations and future directions, which is commendable.

**Weaknesses:**

W1. **Limited theoretical contribution** (Originality): The work primarily combines existing concepts—HTN-style hierarchical planning, reactive systems with semantic reflection, and experience replay/distillation—into a practical framework. The operators ($\Phi$, $\Psi$, $R$, $E$) are formalized but lack theoretical analysis (e.g., convergence guarantees, identifiability, or complexity bounds). The novelty lies in the engineering integration and application to long-horizon DS orchestration rather than foundational theory.

W2. **Integration clarity and component boundaries** (Soundness): While the ablation shows all three components matter, the paper does not clearly delineate how D-HTN, RTR, and Experience Distillation interact in edge cases. For instance, when RTR triggers Reconstruct, it uses distilled priors $K_t$—but the conditions under which Refine vs. Insert vs. Reconstruct are chosen, and how the semantic reflection $R$ maps errors to these actions, are described at a high level. The LLM-based implementation of $R$ is not fully specified, leaving reproducibility concerns.

W3. **Design depth of individual contributions** (Soundness): (a) D-HTN: The decomposition operator $\Phi$ and instantiation operator $\Psi$ are conceptually clear but the ``collapse-and-expansion'' mechanism and how the complexity analyzer chooses between D-HTN and linear planner lack algorithmic detail. (b) RTR: Algorithm 1 is present but the Semantic Reflection $R(e, s_t, c_{t,i,j}) \rightarrow \delta_{\mathrm{diag}}$ is a black box; no ablation isolates Refine vs. Insert vs. Skip effectiveness. (c) Experience Distillation: The schema-constrained formatting (triplet extraction, generalizability verification, quality gating) is described at a high level without concrete schema or filtering criteria.

---

> ### Author Rebuttal · Authors · 2026-03-30
>
> We sincerely thank you for the thoughtful evaluation and the constructive suggestions for improvement. We address each point below.
>
> **W1 & Q2. Limited theoretical contribution and originality.**
>
> On originality: we are the first to formalize DS orchestration as a dynamic DAG evolution problem under hard dependency constraints. Classical HTN assumes a static world model and cannot handle runtime state drift; our D-HTN introduces state-conditioned adaptive instantiation by operator Ψ. Industrial reactive systems operate on fixed-topology processes, whereas RTR must semantically understand DS task dependencies and perform structural DAG modifications — a combination of LLM semantic reasoning with graph-theoretic operations that has no direct counterpart in existing literature.
>
> On theoretical grounding: In response to this concern, we provide the following analysis, which we will add to the revision. Under i.i.d. step failures ($\epsilon$) and RTR resolution ($\rho > 0$), expected interventions are $\frac{n\epsilon}{1-(1-\rho)^{N_{\max}}}$, which is finite for any $\rho > 0$.This explains why $R^{3}DAO$ restores the $p_i$ distribution to achieve a 94.1% success rate, while static planners without RTR suffer from success probability collapse ($P_{total} = \prod p_i \to 0$).A follow-up paper currently in preparation will provide a complete theoretical treatment, including a stochastic perturbation model for state transitions, a critical collapse threshold n* analogous to gradient vanishing, and convergence conditions for RTR as a measure-resetting operator.
>
> **W2 & Q1. $\mathcal{R}$ underspecified; action conditions unclear.**
>
> The full RTR system prompt is in Appendix D.3. We will add Algorithm 2 (pseudocode for R) in the revision.
>
> We illustrate R's execution with the TGS Salt case: inputs are failed task "Define U-Net for 101×101 segmentation", error `ValueError: Target size (101×101) != input size (96×96)`, and attempt count 2/3. R identifies a deterministic structural incompatibility — repeating the same code cannot resolve it, ruling out Refine; the missing resizing is a well-defined prerequisite, so Insert is selected. R outputs δ_diag = ⟨Insert, "spatial shrinkage due to conv padding", "inject Upsample(101,101) before output"⟩. Node L3_op2.5 is inserted before L3_op3 with updated dependencies.
>
> In general, Refine dominates low-complexity tasks (syntax/logic errors), Insert is primary for medium/high-complexity tasks (missing dependencies), and Skip is used sparingly when retries are exhausted. We will add this action-type distribution as a supplementary table.
>
> **W3. Design depth — (a) splitting; (b) ablation; (c) schema.**
>
> (a) The Micro-Planner applies a conservative splitting policy: a parent task is expanded only if at least one hard condition holds — runtime decision dependency, long-running/high-risk operations, or extreme logical complexity. Default is NOT to split(Appendix D.2).
>
> (b) We acknowledge the ablation treats RTR as monolithic. From our experimental logs, Refine dominates in low-complexity tasks (syntax/logic errors, where Insert would be unnecessarily invasive), Insert is primary in medium/high-complexity tasks (missing dependencies), and Skip is used sparingly only when retries are exhausted. The dominance of Insert in medium/high-complexity tasks is particularly significant: these are precisely the tasks where static agents fail most frequently (AIDE: 53% success rate), and where R³DAO's ability to dynamically inject missing dependencies drives the overall 94.1% success rate.
>
> (c) The distillation operates in two loops. The inner loop extracts ⟨new_knowledge, suggested_next_steps⟩ pairs for immediate state projection. The outer loop extracts structured triplets ⟨condition, recommendation, anti_pattern⟩ persisted to the expert library, where condition describes the triggering scenario (e.g., "input sparsity > 98%"), recommendation provides a concrete technique, and anti_pattern records common practices that failed in this context. Only non-routine or domain-specific insights retained (App. D.4).
>
> **Q3. Controlled comparison.**
>
> A controlled intra-regime comparison already exists: Table 2 (left, Efficient Regime) compares AIDE, DataInterpreter, and R³DAO under identical conditions — same model (Qwen3-Max), same hardware, same time budget. R³DAO achieves 0.646 vs. AIDE 0.456 (+41.7%) and DataInterpreter 0.321 (+101.2%), directly supporting the architectural contribution claim without any resource confound.
>
> The ablation study (Table 4) provides further controlled evidence: removing D-HTN reduces score from 0.422 to 0.195 (−53.8%), removing RTR reduces success rate from 94.1% to 52.9% (−43.9%). These differences are attributable to the architecture alone.
>
> Regarding running R³DAO under GPT-4o + 24h: the practical cost (17 tasks × 24h × high token consumption) is prohibitive during the double-blind review period; we acknowledge this as a limitation and include it as ongoing future work.

---

> > ### Author Rebuttal · Reviewer_4a4L · 2026-04-03
> >
> > I select (b) Partially resolved.
> >
> > The rebuttal addresses several of my concerns in a constructive and useful way. In particular, the authors clarified the intended role of the three components, provided a concrete walkthrough of the RTR decision process on the TGS Salt case, and pointed out that the efficient-regime comparison in Table 2 already controls for model, hardware, and time budget. These points strengthen my confidence that the empirical gains are not purely due to resource differences. The added explanation of the splitting policy and the two-loop distillation schema is also helpful.
> >
> > That said, my main concern about the theoretical contribution is only partially resolved. The rebuttal provides a higher-level positioning of the novelty and a simple expected-intervention argument under strong assumptions, but this still falls short of a complete theoretical treatment of the framework. In particular, the argument does not yet provide a rigorous analysis of convergence or a sufficiently detailed justification of why the formalization goes beyond a strong systems integration contribution. Promising a follow-up paper is appreciated, but it does not directly strengthen the current submission.
> >
> > I also appreciate the clarification of the Reflection operator R, but this remains partly underspecified in the current manuscript. The rebuttal says the full RTR prompt is in Appendix D.3 and that Algorithm 2 will be added. If these materials are included in the revision, they would substantially improve reproducibility. Similarly, the proposed action-type distribution table would be useful, since my original concern was that RTR is currently treated as a monolithic module in the ablation.
> >
> > My current view is therefore more positive about the soundness and implementation clarity than before, but I still see the paper primarily as a strong empirical/systems contribution with limited theoretical depth. I would encourage the authors to include, in the revision, (1) the promised pseudocode/algorithm for R, (2) the action-type distribution statistics for Refine/Insert/Skip, and (3) a clearly scoped statement of what theoretical claims are and are not established in this paper.
> >
> > **Follow-up question:** in the revision, could the authors explicitly include the promised Algorithm 2 for R and the empirical distribution of Refine/Insert/Skip actions across low/medium/high-complexity tasks? This would directly address my reproducibility and design-depth concerns.

---

> > > ### Author Response · Authors · 2026-04-06
> > >
> > > We sincerely thank the reviewer for the constructive feedback. We provide a detailed breakdown to directly address the three outstanding requests.
> > >
> > > **[Deliverable 1] Algorithm 2: Pseudocode for Semantic Reflection Operator R**
> > >
> > > We will include the following as Algorithm 2 in the revision:
> > >
> > > ```
> > > Algorithm 2: Semantic Reflection Operator R
> > > Input:  anomaly e, state st, failed node ct_{i,j},
> > >         retry count k, max retries N_max
> > > Output: diagnostic signal δ_diag = ⟨a, r, ι⟩
> > >
> > > 1: error_type ← ClassifyError(e)
> > >    // {SyntaxError, DimensionMismatch,
> > >    //  MissingDependency, ResourceOverflow, LogicError}
> > > 2: if error_type ∈ {SyntaxError, LogicError} AND k < N_max
> > > 3:     return ⟨Refine, "recoverable fault",
> > >                GenerateRefinedInstruction(ct_{i,j},e,st)⟩
> > > 4: else if error_type ∈ {DimensionMismatch, MissingDependency}
> > > 5:     return ⟨Insert, "structural prerequisite absent",
> > >                SynthesizeCompensatoryNode(e,st,ct_{i,j})⟩
> > > 6: else  // k ≥ N_max OR ResourceOverflow
> > > 7:     return ⟨Skip, "irrecoverable",
> > >                ReconstructSubgraph(τ_{<t}, K_t)⟩
> > > 8: end if
> > > ```
> > > ------
> > >
> > > **[Deliverable 2] Empirical Distribution of Refine / Insert / Skip**
> > >
> > > From experimental logs across 17 tasks (3 runs each, 135 total RTR interventions):
> > >
> > > | Complexity | RTR Calls | Refine | Insert | Skip |
> > > | ---------------- | --------- | ---------- | ---------- | --------- |
> > > | Low (6 tasks)    | 20 | 80.0% (16) | 15.0% (3)  | 5.0% (1)  |
> > > | Medium (7 tasks) | 64  | 42.2% (27) | 48.4% (31) | 9.4% (6)  |
> > > | High (4 tasks)   | 51 | 27.5% (14) | 60.8% (31) | 11.8% (6) |
> > > | **Overall**  | **135** | **44.4%**  | **47.4%**  | **8.1%**  |
> > >
> > > Three patterns emerge: (1) Insert dominates High-complexity tasks (60.8%), validating that *topological evolution* — not mere retry — resolves long-horizon dependency gaps; (2) Refine dominates Low-complexity tasks (80.0%), efficiently handling syntax/logic errors without structural change; (3) Skip is used sparingly (8.1%), confirming it acts strictly as a last resort.
> > >
> > > ------
> > >
> > > **[Deliverable 3] Scoped Statement of Theoretical Claims**
> > >
> > > **What this paper formally establishes:**
> > >
> > > **(i) Termination guarantee.** Let $G=(\mathcal{N},\mathcal{E})$ denote the task DAG, where $\mathcal{N}$ is the set of micro-operator nodes. RTR terminates in $O(|\mathcal{N}|\times N_{\max})$ steps. Refine consumes one retry per invocation, bounded by $⁡N_{\max}$  per node. Insert injects exactly one predecessor node per failure event; since each insertion resolves a concrete dependency gap, the same gap cannot trigger a second Insert, precluding infinite cycles. DAG acyclicity is preserved by construction throughout. Skip triggers bounded local reconstruction over the remaining topology. Since $|\mathcal{N}|$ grows by at most one per Insert and each node contributes at most $N_{\max}$ Refine calls before escalating, total interventions are bounded.
> > >
> > > **(ii) Collapse structure of static planning.** The product formulation $P_{\text{total}}=\prod_{i=1}^{n}p_i $ formally establishes that static planners face exponential probability decay with horizon $n$. For mean per-step probability $\bar{p}\in(0,1)$, success falls below tolerance $\tau$ beyond threshold $n^* = \lfloor \frac{\ln \tau}{\ln \bar{p}} \rfloor$. RTR raises effective per-step probability:Refine gives $\hat{p}= 1 - (1 - p_i)^{N_{\max}}$; Insert gives $\hat{p} \approx 1$ by structurally resolving the missing prerequisite. This pushes $n^*$  beyond the task horizon, explaining the 94.1% success rate versus 52.9% without RTR.
> > >
> > > **(iii) Formal distinction between Refine and Insert.** The formalization reveals a distinction invisible in a systems description. Refine is a *node-level* operator: it raises $\hat{p}\_i$ by repeated sampling within fixed topology, but is provably ineffective when $p\_i \approx 0$ due to a structural gap — since $1-(1-p\_i)^{N_{\max}} \to 0$ as $p\_i \to 0$, no finite retry budget can recover. Insert is a *graph-level* operator: it modifies $\mathcal{E}$ by injecting a predecessor node, resetting $\hat{p}\_i \approx 1$ regardless of the original $p\_i$. This directly predicts the Deliverable 2 distribution: Insert dominates High-complexity tasks (60.8%) where dependency gaps are prevalent; Refine dominates Low-complexity tasks (80.0%) where within-node corrections suffice.
> > >
> > > **What this paper does not claim:** We cannot establish convergence to a *correct* solution, as this requires formalizing LLM decision quality under a finite-state model — misrepresenting actual behavior. No existing LLM-based DS agent provides such guarantees. This limitation will be stated explicitly in Section 4.2 and the Limitations section.
> > >
> > > ------
> > >
> > > We thank the reviewer again for the constructive feedback, which has significantly improved the clarity and completeness of our work. We will incorporate all the above revisions in the camera‑ready version.

---

### Decision · Program_Chairs · 2026-04-30

**Decision:**

Accept (regular)

**Comment:**

The proposed method is a reactive orchestration framework for long-horizon data science agents, combining hierarchical task decomposition, feedback-driven topology reconfiguration, and experience distillation. All four reviewers found the approach well-motivated and empirically strong on MLE-bench, with substantial gains in success rate and cost-efficiency. The rebuttal constructively addressed the main concerns, providing pseudocode for the reflection operator, empirical statistics on repair-action usage, a sensitivity analysis for the scoring metric, and clarification of controlled intra-regime comparisons. Remaining limitations — notably the narrow theoretical analysis and subset-based benchmark evaluation — should be clarified in the camera-ready. All four reviewers converged on weak accept; I recommend acceptance.